# Does digital economy development affect urban environment quality: Evidence from 285 cities in China

**Hao Li** **\*, Zihan Yang**

School of Business, Xinyang Normal University, Xinyang, China

\* xylihao@xunu.edu.cn

**Data Availability Statement:** The basic data used in this paper are basically derived from public information such as China Statistical Yearbook, China Environmental Statistics Yearbook, China Science and Technology Statistical Yearbook, China Education Statistics Yearbook, China High-

## Abstract

The impact of the digital economy (DE) on urban environmental quality (EQ) is a critical aspect of China's economic development. This study investigates the impact of DI on urban EQ using the data from prefecture-level cities spanning the period from 2011 to 2021 and updates some disparate conclusions of related studies. It is discovered that a non-linear correlation exists between DE and urban EQ. Currently, DE can effectively improve local city EQ. This conclusion remains valid even after robustness tests and endogeneity treatment. The impact of DE on improving EQ can be classified as the impact of technological innovation, industrial upgrading, resource allocation, infrastructure construction, environmental governance, and changes in public lifestyle. Heterogeneity analysis reveals that the influence of DE is particularly pronounced in cities located in central and eastern regions of China, those with higher levels of administrative management, resource-based urban areas, and those with more stringent environmental regulations.

## 1. Introduction

Cities are important centers for a variety of resources and factors and key drivers of economic growth. Nevertheless, environmental pollution greatly reduces the quality of urban construction and economic development, and poses considerable threats to the health of residents, transportation safety and climate change (Yuan et al, 2020) [1]. As the world's most populous country and largest emerging economy, China also faces serious environmental pollution problems, cities in particular. For example, according to the China Ecological and Environmental Status Bulletin 2022, the air quality in Chinese cities only reached 64.3% of the World Health Organization's requirements [2]. Environmental pollution has seriously hindered the sustainable development of China's economy and caused huge economic losses (Qi et al., 2022) [3]. China has actively taken various measures to improve the quality of urban environment (Liang and Yang, 2019) [4]. For instance, the Fourteenth Five-Year Plan for National Economic and Social Development of the People's Republic of China proposes to "widely establish sustainable modes of production and lifestyles, significantly improve the ecological environment while minimizing carbon emissions, and basically achieve the goal of building a

tech Industry Statistical Yearbook, China Torch Statistical Yearbook, China Industrial Statistical Yearbook, China Information Industry Yearbook, China Tertiary Industry Statistical Yearbook, New China 60 Years Statistical Data Compilation, China Software and Information Service Industry Development Report, WIND database, and the statistical yearbooks of various provinces and cities in the past years, etc. A few missing data were estimated by interpolation, regression analysis and other methods. Public information can be found from the China Knowledge Network China's economic and social big data research platform at https://data.cnki.net/yearBook/single?id=N2022010277. In addition, the China Statistical Yearbook can be found on the official website of the National Bureau of Statistics of China at http://www.stats.gov.cn/tjsj/ndsj/. Provincial and municipal statistical yearbooks can be found on the official websites of provincial and municipal statistical bureaus, with the exception of Xinjiang, the links to other provinces, cities and autonomous regions areas follows: China Statistical Yearbook: https://data.cnki.net/yearBook/single?id=N2022110021. China Industrial Statistical Yearbook: https://data.cnki.net/yearBook/single?id=N2022010304. China Science and Technology Statistical Yearbook: https://data.cnki.net/yearBook/single?id=N2022010277. China Population and Employment Statistics Yearbook: https://data.cnki.net/yearBook/single?id=N2022040097. China Environmental Statistical Yearbook: https://data.cnki.net/yearBook/single?id=N2022030234. China Energy Statistics Yearbook: https://data.cnki.net/yearBook/single?id=N2022030234 New China 60 years of statistical data compilation: https://data.cnki.net/yearBook/single?id=N2010042091. China Trade and Foreign Economic Statistics Yearbook: https://data.cnki.net/yearBook/single?id=N2022010261 China Labor Statistics Yearbook: https://data.cnki.net/yearBook/single?id=N2022020102. China Financial Yearbook: https://data.cnki.net/yearBook/single?id=N2022040012 China Regional Economic Statistics Yearbook: https://data.cnki.net/yearBook/single?id=N2015070200 China Science and Technology Statistical Yearbook: https://data.cnki.net/yearBook/single?nav=%E6%95%99%E8%82%B2%E7%A7%91%E6%8A%80&id=N2023030111 China Education Statistics Yearbook: https://data.cnki.net/yearBook/single?nav=%E6%95%99%E8%82%B2%E7%A7%91%E6%8A%80&id=N2023030114 China Software and Information Service Industry Development Report: https://data.cnki.net/yearBook/single?nav=%E6%95%99%E8%82%B2%E7%A7%91%E6%8A%80&id=N2022070017 China High-tech

beautiful China". To this end, China strongly advocates the adoption of new environmental economic models, such as the digital economy (DE), to improve the urban environment quality (EQ) through the promotion of green and low-carbon lifestyles.

DE takes data resources as the key element, modern information networks as the main carrier, and the integration and application of information and communication technologies as well as the digital transformation of all elements as the important driving force (Ministry of Industry and Information Technology, 2021). With its rapid development speed, wide scope of radiation and deep influence, DE plays an integral role in reorganizing global factor resources, reshaping the structure of the global economy, and changing the pattern of global competition, through promoting profound changes in the modes of production, life and governance (Wu and Yu, 2022) [5]. In recent years, Chinese policymakers have increasingly recognized the significance of developing DE. However, with the rapid development of DE and the deteriorating ecological environment, academics and industry have turned their attention to the study of the relationship between DE and EQ (Wan and Shi, 2022; Ma et al. 2023) [6, 7]. Some of these studies have used the broadband China policy (Zou and Pan, 2023) [8], information and communication technology (ICT) devices (Huong and Thanh, 2022) [9], and digital transportation infrastructures (Zhang et al., 2023a) [10] as representatives of the development of DE, and investigated their effects on $SO_2$ (Wan and Shi, 2022) [6], $CO_2$ (Zhang et al., 2023a) [10], haze pollution (Che and Wang, 2022) [11], urban water pollution, gas pollution, powder (smoke) dust pollution (Bai et al., 2022) [12] and other specific pollutants. DE can promote technological innovation (Zhao et al.,2023) [13], industrial structure upgrading (Wu and Sao, 2022) [14], resource allocation (Qian et al., 2021) [15], digital technology infrastructure construction (Zhang et al., 2023a) [10], clean energy use (Chen, 2022) [16], environmental management performance improvement (Hu et al., 2023) [17], low-carbon green travel (Zhang et al., 2023b) [18], thus improving environmental supervision mechanism (Chang et al., 2022) [19] and enhancing EQ. Nonetheless, most existing literature fails to comprehensively assess the integrated level of DE and the integrated level of EQ by constructing a comprehensive index system, let alone thoroughly study how DE affects the integrated urban EQ. If environmental pollution is directly regarded as urban EQ, it will lead to a lack of representativeness in the research results (Li et al., 2014) [20]. In fact, the comprehensive index is crucial to accurately assessing the impact of DE on urban EQ (Zhang et al., 2023a) [10]. What's more, there are still controversies about the impact of DE on EQ. In contrast to studies that argue that the development of DE can effectively improve EQ, some studies argue that DE aggravates the deterioration of EQ instead (Tang and Yang, 2023) [21]. For example, in the process of DE development, infrastructure production and construction require a large number of physical resources, which will increase carbon dioxide emissions (Zhang et al., 2023a) [10]. The significant investment in infrastructure construction and the rapid growth of Internet usage result in increased power and energy consumption during DE development (Aldo et al., 2023) [22]. In addition, some studies note that a non-linear correlation exists between the growth of DE and EQ. The impact of digital technology on environmental sustainability can be both beneficial and harmful (Shahbaz et al., 2021) [23]. Specifically speaking, the proliferation of Internet economy leads to an upsurge of pollutant emission during its initial stage, followed by a decline in the subsequent stage (Higón et al., 2017) [24]. Furthermore, it is observed that digital fundamentals can damage EQ, but when it exceeds a specific threshold, it can bring environmental benefits (Chen et al., 2019) [16].

This study aims to investigate the impact of DE on urban EQ using panel data from 285 Chinese cities from 2011 to 2021. First, as a developing country with the largest and fastest growing economy in the world, China plays an important role in the economic development of other countries in the world in terms of how to effectively deal with the issues between

Industry Statistical Yearbook: https://data.cnki.net/yearBook/single?nav=%E6%95%99%E8%82%B2%E7%A7%91%E6%8A%80&id=N2023030112 In addition, the China Statistical Yearbook can be accessed on the official website of the National Bureau of Statistics of China at http://www.stats.gov.cn/tjsj/ndsj/. China Torch Statistical Yearbook: https://data.cnki.net/yearBook/single?nav=%E6%95%99%E8%82%B2%E7%A7%91%E6%8A%80&id=N2023030089. China Software and Information Service Industry Development Report: https://data.cnki.net/yearBook/single?nav=%E6%95%99%E8%82%B2%E7%A7%91%E6%8A%80&id=N2022070017 Provincial and municipal statistical yearbooks can be found on the official websites of provincial and municipal statistical bureaus, with the exception of Xinjiang, the links to other provinces, cities and autonomous regions are as follows: Beijing Statistical Yearbook: http://tjj.beijing.gov.cn/tjsj_31433/ Tianjin StatisticalYearbook: https://stats.tj.gov.cn/tjsj_52032/tjnj/ Hebei Economic Yearbook: http://www.hetj.gov.cn/hetj/tjsj/jjnj/ Shanxi Statistical Yearbook: http://tjj.shanxi.gov.cn/tjsj/ Inner Mongolia Statistical Yearbook: http://tj.nmg.gov.cn/datashow/pubmgr/publishmanage.htm?m=queryPubData&procode=0003 Liaoning Statistical Yearbook: https://tjj.ln.gov.cn/tjj/tjxx/xxcx/ Jilin Statistical Yearbook: http://tjj.jl.gov.cn/tjsj/tjnj/ Heilongjiang Statistical Yearbook: http://tjj.hlj.gov.cn/tjj/c106782/common_zfxxgk.shtml Shanghai Statistical Yearbook: https://tjj.sh.gov.cn/tjnj/index.html Jiangsu Statistical Yearbook: http://tj.jiangsu.gov.cn/col/col87172/index.html Zhejiang Statistical Yearbook: http://tjj.zj.gov.cn/col/col1525563/index.html Anhui Statistical Yearbook: http://tjj.ah.gov.cn/ssah/qwfbjd/tjnj/index.html Fujian Statistical Yearbook: https://tjj.fujian.gov.cn/xxgk/ndsj/ Jiangxi Statistical Yearbook: http://tjj.jiangxi.gov.cn/col/col38595/index.html Shandong StatisticalYearbook: http://tjj.shandong.gov.cn/col/col6279/index.html Henan Statistical Yearbook: https://tjj.henan.gov.cn/tjfw/tjcbw/tjnj/ Hubei Statistical Yearbook: http://tjj.hubei.gov.cn/tjsj/sjkscx/tjnj/qstjnj/ Hunan Statistical Yearbook: http://tjj.hunan.gov.cn/hntj/tjsj/tjnj/index.html Guangdong Statistical Yearbook: http://stats.gd.gov.cn/gdtjnj/ Guangxi Statistical Yearbook: http://tjj.gxzf.gov.cn/tjsj/tjnj/ Hainan Statistical Yearbook: http://stats.hainan.gov.cn/tjj/tjsu/ndsj/ Chongqing Statistical Yearbook: http://tjj.cq.gov.cn/zwgk_233/tjnj/ Sichuan Statistical Yearbook: http://tjj.sc.gov.cn/scstjj/c105855/nj.shtml Guizhou Statistical Yearbook: http://stjj.guizhou.gov.cn/tjsj_35719/sjcx_35720/gztjnj_40112/tjnj2018/ Yunnan Statistical Yearbook: http://stats.yn.gov.cn/tjsj/tjnj/index.html Shaanxi Statistical Yearbook: http://tjj.

economic development and the environment (Zhang et al., 2023a) [10]. Second, due to China's large population, its transitional dependence on resource environment and technological resources, and its previous crude economic development model, China faces major challenges in improving EQ (Zhao et al., 2023) [25]. Third, DE, as a more advanced economic form after the agricultural and industrial economies, is a key driver of modern economic development and an essential factor in national competitiveness (Wu and Yu, 2023) [5]. At the same time, with green development as the development goal of China's economy, it is of great research significance to explore the relationship between DE and EQ.

This study firstly constructs a comprehensive index evaluation system for DE and urban EQ respectively, and measures them using an objective weighting method. The study shows that DE has developed rapidly in all Chinese cities in recent years, and that urban EQ fluctuates and rises. Secondly, city and year fixed effect models and spatial Durbin models are used to identify the impact of DE on urban EQ. The results show that the development of urban DE and that of urban EQ in China are U-shaped correlated. The development of urban DE exceeds the inflection point of the curve, which is favorable to the improvement of urban EQ. However, the study results of spatial econometric models indicate that there is currently no spatial spillover effect of DE on improving EQ. Then, the analysis of the impact mechanism shows that green technology innovation, industrial structure upgrading, resource allocation optimization, environmental governance, infrastructure construction and changes in public living habits are all important channels through which DE improves urban EQ. Finally, this study also conducts a heterogeneity analysis to examine the differences in the impacts of DE in terms of urban location, administrative level, policy support for DE development, type of urban development and level of environmental regulation. The findings suggest that the positive impact of DE on urban EQ is mainly found in cities in eastern and central China, cities with higher administrative levels, cities with policy support for DE, resource-based as well as non-resource-based cities, and cities with higher levels of environmental regulation.

This study makes four significant contributions. First, this study uses city level data to examine the relationship between economy and urban EQ. In fact, compared with provincial data, municipal data better illustrates the relationship between DE and EQ in Chinese cities. Second, it expands our understanding of the relationship between DE and EQ in China. In previous studies, DE mainly focuses on a particular aspect of digital technology, such as information and communication technology, Internet technology, digital transmission, broadband China, smart China, or digital transportation policy. On the other hand, in terms of EQ indices, existing literature mainly centers on analyzing the emission reduction of exhaust gas, wastewater, and solid waste, with a focus on air pollution, the impact of PM2.5 or carbon dioxide emissions in particular, thereby lacking comprehensive consideration of urban EQ. These existing studies have not comprehensively assessed the differences in the development of DE among Chinese cities or its impact on the comprehensive urban EQ. Against this background, this study constructs a comprehensive index of DE from four dimensions: digital industry development, digital innovation capacity, digital application degree, and digital governance (Zhang et al., 2022a) [26]. This study also fully considers the self-purification ability of the ecological environment, and establishes a comprehensive index evaluation system for urban EQ from the dimensions of environmental pollutant emission and environmental pollutant absorption (Li et al., 2014) [20]. The index system considers industrial and living pollution as the main sources of emissions, and takes account of the effects of natural environmental absorption and anthropogenic environmental management in the meantime. Third, previous studies on the impact path of DE on EQ mainly focus on technological innovation, industrial structure upgrading and environmental governance, ignoring the importance of resource allocation optimization, changes in living habits and infrastructure construction. Therefore, this

shaanxi.gov.cn/tjsj/ndsj/tjnj/ Gansu Statistical Yearbook: http://tjj.gansu.gov.cn/tjj/c109464/info_disp.shtml Qinghai Statistical Yearbook: http://tjj.qinghai.gov.cn/tjData/qhtjnj/ Ningxia Statistical Yearbook: http://nxdata.com.cn/publish.htm?m=getMorePublish&bc=A01&cn=G01.

**Funding:** This study was funded by the Soft Science Research Program of Henan Province under the project 20231466, awarded to HL. Decision Research Bidding Project of Henan Province under the project 2023JC011, awarded to HL. Key Research Projects for Higher Education Institutions of Henan Province under the project 24B790023, awarded to HL. 2023 Training Program for Young Backbone Teachers of Xinyang Normal University, awarded to HL. Research Topics of the Federation of Social Sciences of Henan Province under the project SKL-2023-567, awarded to HL. Postgraduate Education Reform and Quality Improvement Project of Henan Province under the project YJS2022JD30, awarded to HL and ZY. The funders had no role in study design, data collection and analysis, decision to publish, or preparation of the manuscript.

**Competing interests:** The authors have declared that no competing interests exist.

study comprehensively examines the indirect impact mechanism of DE on the improvement of urban EQ. Fourth, in addition to considering the differences in the impact of DE on urban EQ from the perspective of urban location, this study also comprehensively considers the differences in the impact of DE on urban EQ in China by taking into account the heterogeneous impacts of the administrative level of Chinese cities, the policy support for the development of DE, the type of urban development and the intensity of urban environmental governance.

The rest of this study is organized as follows. Section 2 outlines the theoretical analysis and research hypotheses. Section 3 presents the measurement level of DE and EQ development in cities across China. Section 4 illustrates the research design, which includes sample and data, variables, and methods. Section 5 demonstrates the empirical results. The last section provides a summary of the conclusions and policy implications.

## 2. Theoretical analysis and hypothesis development

### 2.1 DE and urban EQ

The term "urban EQ" refers to the level of appropriateness of the urban environment as a whole or for specific components, such as supporting the survival and growth of both the population and socio-economic activities. It is an environmental assessment concept that captures the particular demands of human beings (Li et al., 2014) [20]. DE is a significant driver of economic growth, which can impact urban EQ through its empowering effects using digital elements, such as technological innovation (Pu et al., 2023) [27], upgrading of industrial structure (Wu and Shao, 2022) [14], allocation of resources (Qian et al., 2021) [15], environmental governance (Chen and Wang, 2022) [11], infrastructure construction (Zhang et al., 2023a) [10] and changing public living habits (Li et al., 2020).

**2.1.1 Technological innovation effect.** DE comprises economic resources generated by digital technological innovation, with technological innovation being the key driver of green development enabled by DE. Rapidly evolving and iterative digital technologies increase the prevalence of knowledge acquisition and dissemination channels, reducing the costs associated with information verification and search (Pu et al., 2023) [27]. Consequently, fragmented information becomes effectively aggregated, thus generating an information spillover effect. The optimal integration and precise matching of components enhances information transmission efficiency, mitigates information asymmetry, fosters openness and transparency in the market, provides a favorable ecosystem for enterprise technological advancements, and derives a number of new digital environmental protection technologies (Li et al., 2019) [28]. Green technology innovation has the potential to enhance the urban EQ through advancements in energy management, pollution control, technology-dependent industries, and the transformation of the innovation landscape (Hu, et al., 2023) [17].

**2.1.2 Industrial structure upgrading effect.** DE boasts numerous advantages, including openness, collaboration, sharing, and connection. It not only transforms the traditional industrial production process and enhances inter-industrial technology penetration, sharing, and integration, but also enables the reconfiguration and optimization of the labor force and other resource elements (Zhang et al., 2019) [29]. The widespread adoption of digital technology enables enterprises to fully utilize data elements, propelling the growth of countless emerging industries, such as big data and artificial intelligence. This fosters digital transformation within industries and accelerates the creation and expansion of new business models (Tang and Yang, 2023) [21]. Effective ways of promoting the green development of related industries include eliminating backward and low-end industries, promoting the industrialization of scientific and technological achievements in environmental protection, encouraging enterprises to digitize, network, and transform intelligently, and facilitating the creation and development

of new business models (Chen, 2022) [16]. Meanwhile, strategic emerging sectors will be promoted, with new industries utilizing non-polluting and clean production factors, ultimately reducing emission of pollutants, such as sulfur dioxide and soot in industrial processes (Aldieri and Vinci., 2022) [22].

**2.1.3 Resource allocation effect.** In the present era of DE, data elements have emerged as the most critical production factors owing to their replicability, widespread availability, and low accessibility costs (Higón et al., 2017) [24]. Arguably, these elements overcome the drawbacks of traditional generating factors, which exhibit diminishing marginal returns (Wu and Shao, 2022) [14]. By reconfiguring labor, capital and technology, digital technology boosts the efficiency of resource utilization and economic cycle. It steers the development of the industrial structure towards high-end green solutions, reduces emissions from production, and effectively addresses the problem of asymmetric information that causes uneven resource allocation (Chen, 2022) [16]. Meanwhile, as a novel economic model of sharing and platform economies, DE utilizes cloud computing and data mining technologies to cater to market demands and consumer preferences, decreases search costs and resource wastage, enhances the sharing of information between supply and demand, refines price matching mechanisms, boosts the efficiency of resource allocation, and efficiently reduces pollutant emission (Qian et al., 2022) [15].

**2.1.4 Environmental governance effect.** Digital technology enhances the means of the environmental monitoring and supervision system and provides a foundation for environmental supervisory bodies to gather environmental information (Ma et al., 2023) [7]. This leads to effective management of environmental pollutant emission, improves the efficiency of governmental environmental supervision and monitoring, and allows for the stipulation of practical environmental policies for green growth (Huong and Thanh, 2022) [9]. Simultaneously, through the digital government's establishment and progression, it is possible for the government to disseminate environmental protection information, comprehend the environmental requirements of businesses and society, provide public access to environmental information, foster an interactive and intelligent setting for communication, and widen the scope for the involvement of primary social agents in environmental administration (Shahbaz et al., 2021) [23]. This promotes the development of informal environmental regulation, with a networked public as its primary entity (Zhao et al., 2023c) [30].

**2.1.5 Infrastructure construction effect.** A well-functioning digital infrastructure is a prerequisite for sustained economic development and prosperity (Qian et al., 2021) [15]. The digital advancement of industries through green transformation and digital industrialization propels the collaborative development of software and hardware in diverse application scenarios, including artificial intelligence, data centers, cloud computing and smart healthcare (Zhang et al., 2023a; Qiao et al., 2022) [10, 31]. This advancement also stimulates the construction of new digital infrastructure, and improves the structure of capital investment, thus resulting in the betterment of urban EQ (Zou and Pan, 2023) [8]. Perfect digital infrastructure can facilitate low-carbon, green travel for individuals, and enhance their low-carbon lifestyle (Wu et al., 2023) [32]. Furthermore, it can enhance the government's environmental management, resulting in more effectiveness and abundance and promoting the "digital pollution reduction dividend" more strongly (Luo, 2020) [33]. While certain research highlights the prospect of digital infrastructure construction leading to substantial consumption of resources and severe environmental pollution, such negative impacts remain confined to the initial phase of infrastructure construction (Higón et al., 2017) [24]. As the infrastructure advances, the cleaner production effect of digital technology progressively counters the broader adverse effect of large-scale infrastructure construction (Aldo et al., 2023) [34].

**2.1.6 Public living habit effect.** The demand for protecting the environment from the public remains to be the primary driving force behind environmental protection. The concept of green consumption gains wide popularity through the Internet (Li et al., 2020) [28]. Green digital consumption platforms, including online shopping, cashless transactions, sharing economy, paper-free office, new businesses, online consumption, new retail, one-stop service, videoconferencing, public transportation, online education, Internet+ and other emerging sectors, subtly reshape consumers' buying habits and actively promote the growth of eco-friendly consumption (Chang et al., 2022; Zhao et al., 2023b) [16, 25]. The adoption of green consumption habits positively alters public lifestyles to match their personalized demands and encourages resource-efficient practices (Zhang et al., 2022) [26]. This will help to form and advocate an eco-friendly and low-carbon way of life. As asserted by Qian et al (2021), the rise of green consumption demands necessitates eco-friendly transformations in production techniques and supply systems [15]. For example, DE makes shared bicycles widely used, thus reducing the frequency of private cars as well as energy consumption and haze pollution (Li et al. 2020) [28].

In summary, this study presents the research hypothesis:

**H1**. The development of DE has the potential to enhance urban EQ through the effects of technological innovation, industrial structure upgrading, resource allocation, environmental governance, infrastructure construction, and the cultivation of green living habits.

## 2.2 Heterogeneous impact of DE on urban EQ

Chinese cities vary in nature, economy, politics and culture, among other aspects. These factors can influence EQ of cities. Is there significant heterogeneity in how DE impacts urban EQ in different Chinese cities? Therefore, this study explores the effect of heterogeneity from various perspectives, including geographic attributes, administrative level of cities, policy support for DE development, and environmental control efforts.

**2.2.1 Geographic attribute level.** The urban DE development in China is likely to have varying effects on different regions' urban EQ. On the one hand, local governments in western China face stronger economic growth pressures and consequently a stronger desire to catch up than those in east-central China (Wan and Shi, 2021) [6]. The drive to catch up leads to a noteworthy rise in economic investment. Digital technology is also more heavily employed in the economic sector. However, the crude economic growth may jeopardize their urban EQ. On the contrary, the level of economic development, DE, and EQ are higher in the eastern and central regions than in the western regions, which may facilitate the utilization of green economic effects of digital technology and consequently maintain a superior EQ (Zhang et al., 2023a) [10]. Therefore, this study proposes the following hypothesis:

**H2a:** The impact of DE on EQ is more significant in eastern and central cities in China.

**2.2.2 Municipal administrative levels.** Municipal administrative levels represent the number of resources that a city has, such as technology, labor and capital, and are a hierarchical expression of its authority to allocate resources (Hua et al., 2018) [35]. Chinese cities are roughly divided into municipalities, provincial capitals, prefectures and counties, according to the administrative level from highest to lowest. In China, an asymmetrical diversity exists between large-scale and small-scale cities, with the former possessing improved healthcare and education facilities (Wan and Shi, 2021) [6], superior infrastructure resources (Qiao et al, 2022) [31], better job prospects and higher wages (Youssef et al., 2021) [36], which contributes

to a "siphon effect". The "siphon effect" results in large urban areas accumulating a plethora of scarce resources, including capital, talent and technology, while the growth of small and medium-sized cities is restricted by a relative lack of resource endowment, the loss of factor inputs, and resource and environmental limitations (Tang and Wang, 2023b) [37]. Cities with higher administrative levels exhibit relatively higher digital economic development and have higher requirements for urban EQ. Therefore, they are more inclined to employ new digital technologies to enhance urban EQ. In view of this, the following hypothesis is presented in this study.

**H2b**: The enhancement impact of metropolitan DE on urban EQ is more noticeable in cities with elevated administrative levels.

**2.2.3 Policy support for DE development.** As a significant driving force for China's economy, policy support for DE development is provided by the Chinese government via establishing pilot digital cities in the early stage of its growth (Qian et al., 2021) [15]. These pilot cities are involved in the construction of broadband infrastructure under the 'Broadband China' policy and the development of 'smart cities' (Zou and Pan, 2023; Wu et al., 2023) [8, 32]. Thanks to the government's policy and financial support, digital infrastructure in cities improves greatly (Zhao et al., 2023a) [13]. This enables DE to have a greater impact, contributing to the sustainable development of the city. In light of this context, the following hypothesis is proposed in this study.

**H2c:** Greater levels of DE development are more effective in enhancing urban EQ.

**2.2.4 Urban development types.** Various aspects of urban technological progress and development models are influenced by levels of resource endowment. The rough development pattern of high energy consumption and high emissions has been utilized by resource-based cities for a long time, leading to excessive carbon emissions and a lack of sustainable development (Zhang et al., 2023a) [10]. Therefore, resource endowments may lead to differences in the effect of DE on urban EQ (Qian et al., 2021) [15].

**H2d:** DE may affect urban EQ differently based on cities' developmental characteristics.

**2.2.5 Urban EQ control efforts.** In order to enhance the scientific decision-making of environmental surveillance and ensure precise and efficient oversight, the government can facilitate the transition of environmental monitoring from conventional manual approaches to contemporary information technology, and from traditional reactive supervision to comprehensive and proactive supervision by implementing digital technology (Che and Wang, 2022) [11]. The effectiveness of environmental regulations is closely linked to the approach to urban economic development. Cities equipped with more stringent environmental regulations will compel their neighboring cities to enhance their production and upgrade their technological capabilities or encourage the relocation of excessively polluting industries, thus consisting with the regional EQ (Liang and Yang, 2019) [4]. In contrast, certain cities with lower economic development and weaker environmental regulations may draw highly polluting enterprises to relocate due to lenient environmental policies (Huong et al., 2022) [9]. Accordingly, this study proposes the following hypothesis:

**H2e:** In urban areas subject to more stringent environmental regulations, DE may prove more effective in enhancing urban EQ.

## 2.3 Spatial spillover impact of DE on urban EQ

DE reduces spatial and temporal distances through efficient information transmission, which is an essential characteristic (Hu et al., 2023) [17]. Due to the spatiotemporal compression effect, DE can expand the breadth and depth of inter-regional economic activity linkages (Zhao et al., 2023c) [30]. Digital technologies have a significant impact on energy consumption, environmental degradation and carbon emissions, while also generating economic spillover effects on a regional scale (Xue et al. 2022) [38]. The rapid development of DE in cities generates scale and agglomeration effects with corresponding incentives. Consequently, such cities become more appealing than their counterparts in surrounding areas, resulting in a "siphon effect" on the resources of surrounding cities (Chen et al., 2020) [39]. Additionally, the effect of DE development in each city may be restricted to the local area on account of the current disparities in development planning and the size of DE.

**H3:** DE development has an uncertain effect on EQ of neighboring cities, and may result in spatial spillover effect, policy siphon effect or no significant spatial effect.

# 3 Measurement level of DE and EQ in cities

Accurately measuring the level of DE and EQ in Chinese cities is crucial for a comprehensive and objective evaluation of the impact of DE on EQ. In this section, we aim to develop a comprehensive index system for urban DE and EQ.

## 3.1 Index system for Chinese urban DE

This study defines DE as encompassing all economic activities reliant on Internet and communication technologies. Given the multi-sectoral characteristic of DE, this study uses the DE framework devised by the China Academy of Communications (2020) to establish an index system for gauging the extent of advancement in urban DE across four primary dimensions: fundamentals of digital industry development, digital innovation capability, digital application degree, and digital governance (Zhang et al., 2022b) [40] (as shown in Table 1).

**Table 1. Index evaluation system for urban DE.**

| Primary index | Secondary index | Index description | Unit | Index attribute |
|---|---|---|---|---|
| Fundamentals of digital industry development | Telecom industry output | Total telecom services | $10^4$ CNY | + |
| | Development of e-commerce industry | Number of urban e-commerce parks | - | + |
| | Foundation of information industry | Proportion of employees engaged in information transmission, computer services and software | % | + |
| Digital innovation capability | Digital innovation element support | Science and technology expenditure | $10^4$ yuan | + |
| | Digital innovation output level | Number of DE related patents per 10,000 people | - | + |
| | Digital high-tech penetration | Penetration of digital high-tech application in listed companies | - | + |
| Digital application degree | Internet application | Number of Internet users per 100 people | Household | + |
| | Mobile Internet applications | Number of mobile phone users per 10,000 people | Household | + |
| | Application of digital finance | Digital financial digitization index | - | + |
| Digital governance | Digital e-Government Level | Number of government websites | - | + |
| | Number of new government media | Official government accounts on new media platforms | - | + |

In Table 1, "penetration of digital high-tech application in listed companies" mainly reflects the penetration level of high-tech by calculating the frequency of subdivision indices of artificial intelligence technology, blockchain technology, cloud computing technology, big data technology and digital technology application in the reports of listed companies, and then summarizing them as the urban scale on average.

## 3.2 Index system for Chinese urban EQ

The existing selection of environmental pollution indices typically concentrates on industrial pollutant emission. Unfortunately, this approach fails to offer comprehensive consideration of the sources of environmental pollutant emission (Li et al., 2014) [20]. Furthermore, current research predominantly focuses on selecting fundamental pollution indices from pollution receptors, including the atmosphere, water and soil (Bai et al., 2022) [12]. However, this disregards the adverse effects of human actions and lifestyles on EQ. Additionally, previous studies often equate the emission of pollutants with the extent of environmental contamination, overlooking both humanity's contributions to mitigating environmental issues and the self-cleaning properties of nature concerning environmental pollution (Li et al., 2014) [20]. In this study, we fully consider the self-purification capacity of the ecological environment and develop a comprehensive index evaluation system for urban EQ based on the environmental pollutant emission and environmental pollutant absorption dimensions. The index system considers both industrial and living pollution as the primary emission sources, and takes account of natural environmental absorption and the impact of man-made environmental management at the same time (as shown in Table 2).

## 3.3 Evaluation methods for composite index measurements

This study spans from 2011 to 2021, and covers 285 cities, making it representative and sufficient to accurately reflect the overall situation in China. In addition, we choose 2011 to 2021 as the research period because the digital financial inclusion data started in 2011. It's worth noting that the latest raw data for a city's carbon targets were available until 2019. Therefore, to align the data with 2021, this study utilizes the average annual value added from 2011 to 2019. The selected variables data in this study are gathered from relevant statistical yearbooks of previous years, the WIND database, local statistical bureau websites, statistical research reports, or the Internet. To account for missing data, we employ interpolation methods.

The calculation of the comprehensive indices includes identifying the attributes of the fundamental evaluation indices, establishing the weights of the evaluation indices, and determining their scales. To ensure data continuity, integrity and comparability over time, this study begins by using the data extreme value standardization processing method to standardize raw data based on 2011. Then, the dynamic comprehensive evaluation method is utilized to allocate weights to basic indices. Finally, the corresponding composite indices are calculated using the linear weighting method. The specific calculation is as follows.

(1) Normalization of data for each criterion. As each index has a different dimension and unit, it is necessary to standardize the indices using dimensionless processing to avoid unfairness. Employing dimensionless processing greatly impacts the methods used to draw conclusions, so a dimensionless method is the most suitable approach to ensure that evaluated factors are distinct (Yi et al., 2014) [41]. Due to the variation in magnitude and measurement of data across criteria, both positively and negatively oriented criteria undergo normalization via the

**Table 2. Index evaluation system for urban EQ.**

| Primary index | Secondary index | Index Description | Unit | Index attribute |
|---|---|---|---|---|
| Industrial pollution | Total industrial wastewater discharge | Pressure of socio-economic development on the water environment | Billion tonnes | - |
| | Total industrial dust emissions | The pressure of socio-economic development on the atmospheric environment | Million tonnes | - |
| | Industrial solid waste generation | The pressure of socio-economic development on the ecological environment | Million tonnes | - |
| Living pollution | Total Sulphur dioxide emissions | The pressure of socio-economic development on the atmospheric environment | Million tonnes | - |
| | Amount of living waste removed | Pressure on the ecosystem | Million tonnes | - |
| | Living wastewater discharge | Pressure on the water environment | Million tonnes | - |
| | Environmental Noise Monitoring Equivalence Levels | Human health and living environment | Decibels | - |
| Carbon emissions | Total carbon dioxide emissions | The pressure of socio-economic development on the atmospheric environment | Million tonnes | - |
| Natural absorption | Green space per capita | Living environment and quality of life | Sqm/Person | + |
| | Greening coverage of urban built-up areas | Extent of improvement in pollution through greening | % | + |
| | Average daily temperature | Impact on the removal of atmospheric pollutants | Degrees Celsius | + |
| | Average annual precipitation | Impact on suction removal of pollutants | mm | + |
| Environmental governance | Average urban relative humidity | Impacts on climate | % | + |
| | Water resources per capita | Living environment and quality of life | Billion cubic metres | + |
| | Industrial wastewater compliance discharge | Industrial pollution control | Billion tonnes | + |
| | Industrial dust removal | Industrial pollution control | Million tonnes | + |
| | Integrated Industrial Solid Waste Utilization | Industrial pollution control | Million tonnes | + |
| | Percentage of built-up area in smoke control areas | Environmental construction | % | + |
| | Harmless treatment capacity of living waste | Living pollution control | Million tonnes | + |
| | Living sewage treatment capacity | Living pollution control | Billion tonnes | + |
| | Area share of noise compliance zones | Human health and living environment | % | + |

extreme value method in accordance with Eq (1).

$$s_{ij}(t_k) = \begin{cases} \dfrac{\max[x_j(t_0)] - x_{ij}(t_k)}{\max[x_j(t_0)] - \min[x_j(t_0)]} \\[2ex] \dfrac{x_{ij}(t_k) - \min[x_j(t_0)]}{\max[x_j(t_0)] - \min[x_j(t_0)]} \end{cases} \tag{1}$$

Where $x_{ij}(t_k)$ is the actual value of criterion j for object i, $s_{ij}(t_k)$ is the normalized value, and $\max[x_j(t_0)]$ and $\min[x_j(t_0)]$ are the maximum and minimum value for criterion j, respectively. i = 1, 2, · · ·, m, and j = 1, 2, · · ·, n. Specifically, formula (1) is used for negative indices, and the following formula is used for positive indices.

(2) Dynamic comprehensive evaluation method. This study adopts a dynamic comprehensive evaluation method (DCEM) and integrates different subsystem indices into a single number (Guo, 2007) [42]. The theory is illustrated as below:

For $\{x_{ij}(t_k)\}$, the comprehensive function is arranged by time $t_k(k = 1, 2, \cdots, N)$:

$y_i(t_k) = \sum_{j=1}^{m} \omega_j s_{ij}(t_k), (k = 1, 2, \cdots, n, i = 1, 2, \cdots, m)$, $y_i(t_k)$ is the sum of the evaluation value of the calculated area at $t_k$ and this study includes 11 years, $\omega_j$ is the index weight, and $s_{ij}(t_k)$ is city $i$ with $j$ evaluation standard at $t_k$. This study evaluates 285 cities. $j$ represents the secondary index of the corresponding index system. $\omega_j$ shows the largest difference on the panel level, which is:

$$\sigma^2 = \sum_{k=1}^{N} \sum_{i=1}^{n} \left( y_i(t_k) - \bar{y} \right)^2 \tag{2}$$

A dimensionless process shall be applied to the original data $\{x_{ij}(t_k)\}$, then

$$\bar{y} = \frac{1}{N} \sum_{k=1}^{N} \frac{1}{n} \sum_{i=1}^{n} \sum_{j=1}^{m} \omega_j x_{ij}(t_k) = 0,$$

and Eq (3) can be expressed as:

$$\sigma^2 = \sum_{k=1}^{N} \sum_{i=1}^{n} \left( y_i(t_k) \right)^2 = \sum_{k=1}^{N} \left[ \omega^T H_k \omega \right] = \omega^T \sum_{k=1}^{N} H_k \omega \tag{3}$$

Where $\omega = (\omega_1, \omega_2, \cdots, \omega_m)^T$, and $H = \sum_{k=1}^{N} H_k$ is a $m \times m$ order symmetric matrix:

$$H_k = A_k^T A_k (k = 1, 2, \cdots, N) \text{ and } A_k = \begin{bmatrix} x_{11}(t_k) & \cdots & x_{1m}(t_k) \\ \cdots & & \cdots \\ x_{n1}(t_k) & \cdots & x_{nm}(t_k) \end{bmatrix} \tag{4}$$

The following conclusions can be drawn:

① If $\omega^T \omega = 1$, $\sigma^2$ is maximized $\left( \max\limits_{\|\omega=1\|} \omega^T H \omega = \lambda_{\max}(H) \right)$ when $\omega$ is a feature vector corresponding to the largest value in matrix $H$.

② If $H_k > 0 (k = 1, 2, \cdots, N)$, we use the DCEM in both the horizontal and vertical directions, respectively, at $t_k$, resulting in identical ranking.

(3) Linear weight method. By combining the vector of weight coefficients and the standardized values of indices, it is possible to calculate the composite index of a specific city in a given year using the linear weight method. The calculation formula for this is written as:

$$Y_i(t_k) = \sum_{j-1}^{m} \omega_j s_{ij}(t_k) \tag{5}$$

## 3.4 Analysis of measurement results

The calculated DE and EQ are listed in Table 3. Among these indices, a higher value on the index indicates a greater level of DE and EQ.

EQ in Chinese cities exhibits consistent improvement except for slight declines recorded in 2012, 2015, 2016, and 2019. Eastern region reports the highest level of EQ, followed by the western region, whereas the central region experiences the lowest EQ level.

**Table 3. Development level of urban DE and urban EQ in China.**

| Year | Level of DE development | | | | Comprehensive quality of urban environment | | | |
|---|---|---|---|---|---|---|---|---|
| | Overall China | Eastern region | Central region | Western region | Overall China | Eastern region | Central region | Western region |
| 2011 | 0.1573 | 0.1681 | 0.1479 | 0.1555 | 0.5488 | 0.5344 | 0.5720 | 0.5464 |
| 2012 | 0.1742 | 0.1866 | 0.1667 | 0.1692 | 0.5418 | 0.5333 | 0.5662 | 0.5325 |
| 2013 | 0.2115 | 0.2261 | 0.1997 | 0.2079 | 0.5472 | 0.5418 | 0.5664 | 0.5388 |
| 2014 | 0.2229 | 0.2364 | 0.2175 | 0.2131 | 0.6411 | 0.6358 | 0.6688 | 0.6263 |
| 2015 | 0.2355 | 0.2538 | 0.2259 | 0.2248 | 0.6395 | 0.6344 | 0.6747 | 0.6190 |
| 2016 | 0.2519 | 0.2761 | 0.2398 | 0.2375 | 0.6332 | 0.6244 | 0.6592 | 0.6230 |
| 2017 | 0.2583 | 0.2839 | 0.2477 | 0.2428 | 0.6450 | 0.6319 | 0.6663 | 0.6425 |
| 2018 | 0.2648 | 0.2902 | 0.2541 | 0.2492 | 0.6527 | 0.6406 | 0.6808 | 0.6444 |
| 2019 | 0.2718 | 0.2961 | 0.2617 | 0.2579 | 0.6497 | 0.6387 | 0.6702 | 0.6458 |
| 2020 | 0.2791 | 0.3024 | 0.2704 | 0.2651 | 0.6600 | 0.6516 | 0.6812 | 0.6529 |
| 2021 | 0.2863 | 0.3087 | 0.2781 | 0.2722 | 0.6660 | 0.6554 | 0.6898 | 0.6592 |
| Average | 0.2376 | 0.2571 | 0.2281 | 0.2268 | 0.6204 | 0.6111 | 0.6450 | 0.6119 |

Note: The above indices are the average values of all cities in different regions for each year. Judging from Table 3, it is evident that from 2011 to 2021, China's DE level experiences a steady rise both nationally and regionally, with the index of DE increasing from 0.1573 to 0.2863. The eastern region shows the highest level of DE, while the western region lags behind.

# 4 Research design

## 4.1 Methodology

**4.1.1 Base model.** To examine the impact of DE on urban EQ, this study develops an equation set model within the IPAT model framework. The IPAT model generally analyses the effects of different factors, including population, economy and technology, on the environment (Shen et al., 2018) [43]. This study establishes the equation model for the impact of DE on urban EQ, as shown below:

$$EQ_{it} = \alpha_0 + \alpha_1 DE_{it} + \sum_{j=1}^{n} \beta_j X_{itj} + \mu_i + \lambda_t + \varepsilon_{it} \tag{6}$$

Where $EQ$ denotes urban EQ; $DE$ denotes the level of DE; $X$ comprises the remaining control variables; $\mu_i$ denotes city fixed effects and $\lambda_t$ denotes year fixed effects; and $\varepsilon_{it}$ denotes the random perturbation term.

**4.1.2 Spatial effect model.** Interregional "siphon effect" and "spillover effect" often result in mutual infiltration and radiation of environmental pollution and economic activities amongst cities (Tang and Wang, 2023b) [37]. It is concluded that digital technologies have spatial spillover effect (Zhao et al. 2023) [30]. Ignoring the influence of spatial effects may lead to biased model estimation and a lack of explanation. The spatial Durbin model (SDM) takes into account not only the spatial correlation of the dependent variables but also that of the independent variables, therefore better reflecting the actual situation and more effectively demonstrating the impact of the independent variables on the dependent variables (Wang and Liu, 2023) [44]. Taken together, this study presents a spatial Durbin model for analysis. The model is constructed as follows:

$$EQ_{it} = \rho W EQ_{it-1} + \alpha_0 + \alpha_1 DE_{it} + \alpha_3 X_{ij} + \theta W DE_{it} + \beta W_{ij} X_{ij} + \mu_i + \lambda_t + \varepsilon_{it} \tag{7}$$

Where $EQ_{it-1}$ represents the lagged term of the explanatory variables and focuses primarily on

investigating the dynamic effects of DE on urban EQ; $\rho$ represents the spatially lagged regression coefficient, indicating the extent of mutual influence of EQ among neighboring cities; $\theta$ denotes the spatial regression coefficient of the explanatory variables; and $W$ represents the spatial weighting factor. The current study utilizes a spatial weighting matrix that considers both geographic distance and economic factors.

## 4.2 Variables

The preceding sections detail the dependent variables of urban EQ and the independent variable of DE development. The remaining control variables are primarily listed below.

As per existing literature, this study also incorporates control variables that potentially affect urban EQ (Qi et al., 2022; Zhang et al., 2022a) [3, 26]. Specifically, these control variables consist of economic development, technical level, environmental regulation, industrial structure, foreign investment, urbanization level and energy consumption. Economic development refers to the growth rate of GDP. Technical level is measured by taking the natural logarithm of per capita financial expenditure on science and technology. Environmental regulation is typically expressed as the percentage of investment in urban environmental governance. Industrial structure is determined by the proportion of added value from tertiary industries. Foreign investment is measured as the percentage of GDP that comes from direct foreign investment. Urbanization level refers to the percentage of the year-end population living in households within the city. Energy consumption is determined by dividing coal consumption by total energy consumption. The description statistics for each variable is shown in Table 4.

## 4.3 Spatial correlation analysis

Prior to utilizing the spatial econometric model, an assessment is conducted on the Moran's $I$ to determine whether spatial effects should be taken into consideration in the empirical research. The Moran's $I$ can be expressed as the standardized spatial weight matrix.

$$I = \frac{\sum_{i=1}^{n} \sum_{j=1}^{n} w_{ij}(x_i - \bar{x})(x_j - \bar{x})}{\sum_{i=1}^{n} (x_i - \bar{x})^2} \tag{8}$$

$x_i$ represents the observed value of the initial region i, whereas the spatial weight matrix is given by W. Additionally, the total number of regions is represented by n. Table 5 displays the results of calculating the Moran index for EQ and DE development levels over the years.

**Table 4. Descriptive statistics for each variable.**

| Variable | | Mean | SD | Min | Max |
|---|---|---|---|---|---|
| DE | *DE* | 0.2189 | 0.0049 | 0.051 | 0.5368 |
| EQ | *EQ* | 0.6580 | 0.1788 | 0.3246 | 0.9938 |
| Economic level | *Eco* | 7.1371 | 0.9653 | 3.5541 | 10.3298 |
| Environmental regulation | *Er* | 0.5863 | 0.7918 | 0.034 | 17.83 |
| Industrial structure | *Indu* | 0.5609 | 0.0106 | 0.3381 | 0.9669 |
| Energy consumption | *Ene* | 0.6839 | 0.3955 | 0.2767 | 2.3108 |
| Foreign investment | *Fdi* | 0.021 | 0.029 | 0.000 | 0.3450 |
| Technological level | *Inno* | 4.0744 | 1.6431 | 1.7026 | 9.6803 |
| Urbanization level | *City* | 0.5372 | 0.0935 | 0.3381 | 0.8961 |

**Table 5. Measurement results of Moran's I index.**

| Year | EQ | | DE | |
|---|---|---|---|---|
| | **Moran's *I*** | **Z-value** | **Moran's *I*** | **Z-value** |
| 2010 | 0.243** | 2.139 | 0.233** | 2.157 |
| 2011 | 0.264** | 2.303 | 0.233** | 2.157 |
| 2012 | 0.278** | 2.41 | 0.228** | 2.118 |
| 2013 | 0.274*** | 2.378 | 0.225** | 2.094 |
| 2014 | 0.277** | 2.401 | 0.225** | 2.088 |
| 2015 | 0.249** | 2.191 | 0.224** | 2.079 |
| 2016 | 0.260** | 2.286 | 0.222** | 2.070 |
| 2017 | 0.245** | 2.162 | 0.218** | 2.040 |
| 2018 | 0.241** | 2.221 | 0.223** | 2.120 |
| 2019 | 0.243** | 2.149 | 0.216** | 2.027 |
| 2020 | 0.255** | 2.410 | 0.228** | 2.312 |
| 2021 | 0.249** | 1.991 | 0.246** | 2.217 |

Notes:

* denotes $p < 0.1$,

** denotes $p < 0.05$, and

*** denotes $p < 0.01$.

Table 5 demonstrates a positive global Moran's *I* index for DE and EQ in Chinese cities from 2011 to 2021, passing the 5% level of significance. This indicates a spatial clustering and interdependence that can be further explored through the use of spatial econometric models.

## 5 Results and discussions

### 5.1 Benchmark results

Before conducting regression analysis, this study uses VIF for multicollinearity testing. The results show that all VIF values are smaller than 10, indicating that there is no multicollinearity between variables. This study employs a two-way fixed effects model for testing and presents the regression results in Table 6.

The regression result (1) indicates a significantly positive coefficient of influence between ED and urban EQ, when other control variables are not considered. The outcome suggests that the development of DE can drastically enhance urban EQ. For each unit increase in DE, the urban EQ is enhanced by 0.027 units. This finding is consistent with Ma et al (2023) [7]. The growth of DE has the potential to enhance urban EQ through the effects of technological innovation, upgrading industrial structures, optimizing resource allocation, and the like (Zhang et al., 2023a) [10].

The regression result (2) indicates that even with other control variables considered, DE can effectively enhance urban EQ.

In regression result (3), this study incorporates the quadratic term of DE into the regression model to investigate the non-linear impact of DE on urban EQ. According to the regression results, the primary term of DE and urban EQ has a significantly negative regression coefficient, whilst the secondary term has a significantly positive regression coefficient. It is demonstrated that a positive U-shaped curve exists between DE and urban EQ in Chinese cities. After calculating the inflection point of the U-shaped curve, which is 0.2317, and combining it with the average value of China's DE over the years as displayed in Table 3, it is evident that the

**Table 6. Benchmark regression.**

| | Panel Fixed Effect | | | SYS-GMM | Spatial Durbin | |
|---|---|---|---|---|---|---|
| | **(1)** | **(2)** | **(3)** | **(4)** | **(5)** | |
| L. EQ | | | | 0.764*** | 0.541*** | |
| | | | | (17.65) | (5.69) | |
| DE | 0.027*** | 0.021*** | -0.019*** | -0.026*** | -0.0403*** | -0.052 |
| | (11.34) | (5.65) | (-7.23) | (-3.16) | (-4.92) | (-0.28) |
| DE^2 | | | 0.041*** | 0.061*** | 0.079*** | 0.0561 |
| | | | (8.26) | (5.68) | (6.12) | (0.81) |
| Eco | | 0.292*** | 0.291*** | 0.354*** | 0.452*** | -0.318 |
| | | (6.46) | (6.34) | (3.08) | (4.36) | (-1.15) |
| Er | | 0.015*** | 0.024*** | 0.034*** | 0.031*** | -0.024* |
| | | (4.591) | (4.245) | (3.841) | (4.914) | (1.871) |
| City | | -0.451*** | -0.547*** | -0.353*** | -0.412*** | 0.344*** |
| | | (-3.13) | (-2.83) | (-4.15) | (-3.45) | (4.12) |
| Ene | | -0.241 | -0.018 | -0.023 | -0.040 | -0.112 |
| | | (-1.02) | (-0.51) | (-0.84) | (-1.10) | (-0.54) |
| Inno | | 0.029*** | 0.026*** | 0.016*** | 0.032*** | 0.010 |
| | | (5.17) | (4.78) | (3.23) | (3.62) | (0.42) |
| Indu | | 0.024*** | 0.018** | 0.023*** | 0.221** | 0.263 |
| | | (4.09) | (2.39) | (3.13) | (2.06) | (1.15) |
| Fdi | | 0.034 | 0.024 | 0.024 | 0.041 | 0.121 |
| | | (0.98) | (0.15) | (0.51) | (1.05) | (0.89) |
| rho | | | | | 0.194*** | |
| | | | | | (5.318) | |
| Cons | | -0.245*** | -0.151*** | -0.123*** | -1.074*** | -1.212*** |
| | | (-4.15) | (-5.06) | (-4.34) | (-4.50) | (-4.31) |
| City FE | YES | YES | YES | YES | YES | YES |
| Year FE | YES | YES | YES | YES | YES | YES |
| Obe | 3135 | 3135 | 3135 | 2850 | 2850 | 2850 |

Notes:

***, ** and * represent significance levels of 1%, 5% and 10% respectively; and values in brackets indicate t-values.

level of DE surpasses the inflection point in 2014. Before 2013, the China's urban EQ declines due to the development of DE. Thereafter, with the development of DE, the urban EQ improves. During the initial stages of DE development, the construction of digital infrastructure and the increased energy consumption of digital equipment will lead to a decline in urban EQ (Zou and Pan, 2023) [8]. With advancements in the construction and development of digital technology infrastructure, the improvement effect of DE on urban EQ is increasingly noticeable (Luo, 2020) [33].

Taking regression result (3) as an example, along with other control variables, the regression coefficient between the level of urban economic development and urban EQ is significantly positive. This suggests that China's current economic growth model, with its focus on high-quality economic development, can effectively enhance EQ while maintaining economic growth. Environmental regulation is significantly and positively associated with urban EQ, and increased investment in environmental governance can effectively improve urban EQ. There is a significant negative correlation between urbanization level and urban EQ. The non-

significant regression coefficient between the energy consumption and EQ of urban areas suggests that the increase in energy consumption does not cause any deterioration in the overall EQ. This outcome may be due to the significant improvements in energy efficiency and the use of cleaner energy sources (Chen, 2022) [16]. The level of innovation in green technology is strongly correlated with urban EQ. This indicates that enhancing green technology leads to the advancement of cleaner production technology, thus reducing production consumption and pollutant emission (Hu et al., 2023) [17]. The positive regression coefficient between upgrading industrial structure and urban EQ indicates that urban EQ can be effectively improved through industrial structure upgrading. Traditional companies are faced with the necessity of restructuring energy-intensive operations and gradually modernizing production structures, which benefits urban EQ (Zhang et al., 2023a) [10]. However, the impact of foreign investment on urban EQ remains undefined. This study does not verify the existence of either the "pollution halo" effect or "pollution shelter" effect.

Since the core explanatory variables in this study are the synthetic composite indices for the level of urban DE and the level of urban EQ, using only a panel data fixed effects model for the regression estimation may result in biases. In this study, the SYS-GMM approach is implemented to perform additional regression analysis that addresses the endogeneity issue. Since the SYS-GMM model amalgamates the DIF-GMM and horizontal GMM models, it improves the estimation efficiency when compared to DIF-GMM estimation (Liu and He, 2019) [45]. The SYS-GMM model (3) regression results demonstrate that the degree of DE remains influential in enhancing the urban EQ effect, which verifies the stability of the regression outcomes of the study.

The spatial Durbin model considers not only the correlation of independent variables, but also the correlation between independent and dependent variables in adjacent areas (Zou and Pan, 2023) [8]. Column (4) in Table 6 employs the dynamic spatial Durbin model to investigate the effect of DE level in neighboring cities on the local urban EQ. It is shown that the spatial auto-correlation coefficient of urban EQ is positive. It is evident that a positive spatial spillover effect occurs in urban EQ. In other words, improving the urban EQ in local region can improve the urban EQ in neighboring regions. However, the DE of local cities can improve their own urban EQ, but it does not have a significant impact on the EQ of neighboring cities. It conforms to research hypothesis H3. The spatial spillover effects of urban EQ improvement are not apparent due to inadequate regional linkage and cities' siphon effect (Chen et al., 2020) [39]. The results of the control variable regression indicate that the improvement of local urbanization level can effectively improve the EQ of neighboring cities. The strict local environmental management policies have a negative impact on the EQ of neighboring cities.

## 5.2 Robustness test and endogeneity problem treatment

**5.2.1 Robustness test.** In the above-mentioned section, we calculate the composite index for the development level of urban DE and urban EQ by employing the dynamic comprehensive evaluation method for weightage. Based on the consideration of the robustness of the regression results, we utilize the entropy value method to evaluate the levels of urban DE and EQ. The regression results in columns (1) and (2) of Table 7 demonstrate that the progression of DE can still efficiently enhance urban EQ.

In order to further test the robustness of the regression results and the dynamic impact of DE development on urban EQ, this study deals with urban EQ by lagging behind one period and two periods respectively. Based on the outcomes in columns (3) and (4) of Table 7, the extent of improvement in urban EQ facilitated by DE remains effective. After adjusting to the

**Table 7. Robustness and endogeneity test results.**

|  | NewDig | NewEnv | One Lag | Two Lags | Instrumental Variable | |
| --- | --- | --- | --- | --- | --- | --- |
|  | (1) | (2) | (3) | (4) | (5) | (6) |
| DE | 0.014** | 0.018*** | 0.038*** | 0.032*** | 0.018*** | 0.027*** |
|  | (2.27) | (7.55) | (15.37) | (17.54) | (4.21) | (6.36) |
| Control Variable | YES | YES | YES | YES | YES | YES |
| City FE | YES | YES | YES | YES | YES | YES |
| Year FE | YES | YES | YES | YES | YES | YES |
| LM test | \ | \ | \ | \ | 75.561*** | 121.541*** |
| F-test | \ | \ | \ | \ | 125.451 | 140.512 |
| Obs | 3135 | 3135 | 2850 | 2560 | 2850 | 2850 |

Notes:

***, ** and * represent significance levels of 1%, 5% and 10% respectively; and values in brackets indicate t-values.

delayed influence of DE on EQ, it is apparent that the favorable impact of DE on urban EQ is reinforced. This provides evidence of time lag in DE impact on urban EQ. Moreover, the conclusion of the study again verifies the reliability of the regression analysis.

**5.2.2 Endogeneity problem treatment.** Considering the potential for biased regression outcomes stemming from endogeneity issues, this study employs the cross terms of historical data on post and telecommunications, represented by the number of post offices and telephones per 10,000 people at the end of 1984 in selected cities, together with previous year data on Internet users, as instrumental variables to determine the DE levels of the cities (Zhang et al., 2023a) [10]. Digital technology originates from the promotion of fixed line telephone. Cities with high penetration of fixed line telephone in history generally have a high level of DE development. At the same time, before the popularization of fixed telephone, people use the post office system as the main way of information communication. The post office system is also the executive department of fixed telephone, which affects the construction of fixed telephone, Internet access, and other facilities to a certain extent, and shapes the development of DE, thereby meeting the relevance. In addition, the impact of traditional telecommunication tools, such as fixed line telephone and post office, on current DE development is gradually reduced to meet exclusivity (Zhang et al., 2023a) [10].

After creating the cross terms of instrumental variables and time dummy variables, the second stage regression results based on two-stage least squares method are shown in columns (5) and (6), which demonstrate that the impact of DE on urban EQ remains significant even after accounting for endogeneity. The instrumental variables from the unidentifiable LM test and weak instrumental variables from the F test both confirm the reliability and robustness of the study's findings.

## 5.3 Impact mechanism results

Based on previous estimation results, this study further analyzes the impact mechanisms of DE on urban EQ from the perspective of the technological innovation effect, industrial structure upgrading effect, resource allocation optimization effect, environmental governance effect, capital investment adjustment effect, public living habit change effect, and infrastructure construction effect. The results are presented in Table 8.

**5.3.1 Green technological innovation effect.** The technological innovation effect pertains to the advancement and refinement of industrial development models utilizing digital

**Table 8. Path test of DE to improve urban EQ.**

|  | Inno | Indu | Reso | ER | Infra | Pub |
|---|---|---|---|---|---|---|
|  | (1) | (2) | (3) | (4) | (5) | (6) |
| DG | 0.0124*** | 0.0281*** | 0.0128* | 0.0026*** | 0.0328*** | 0.0114*** |
|  | (4.31) | (3.54) | (1.87) | (8.54) | (6.75) | (4.25) |
| Control variant | YES | YES | YES | YES | YES | YES |
| City FE | YES | YES | YES | YES | YES | YES |
| Year FE | YES | YES | YES | YES | YES | YES |
| Obs | 3135 | 3135 | 3135 | 3135 | 3135 | 3135 |

Notes:

***, ** and * represent significance levels of 1%, 5% and 10% respectively; and values in brackets indicate t-values.

technologies, such as Internet technology, big data analysis and artificial intelligence (Zhao et al., 2023a) [13]. The improvement of technological innovation level can promote the efficient utilization of resources, reduce the pollutant emission generated by production and daily activities, and improve the technology intended for effective pollutant treatment (Shahbaz et al., 2021) [23]. The level of technological innovation is expressed by the number of green invention patents present in each region (Qi et al., 2022) [3].

The results in Column 1 of Table 8 demonstrate that the estimated DE coefficient on technological innovation is 0.012, which is statistically significant at the significance level of 1%. This outcome suggests that urban green technological innovation benefits from the development of DE, which significantly stimulates urban technological innovation. The expansion of DE enriches access to knowledge and fosters a more transparent market environment, creating optimal circumstances for technological innovation. Green technology innovation can enhance urban EQ through advancements in energy management, pollution control, and technology-enabled industries, as well as reshaping the innovation environment (Qian et al., 2021; Hu et al., 2020) [15, 17]. Therefore, the effect of technological innovation is a significant mechanism for DE to enhance urban EQ.

**5.3.2 Industrial structure upgrading effect.** The effect of upgrading industrial structure entails the elimination of highly polluting and energy-intensive industries, as well as the acceleration of strategic emerging and modern service industries (Wu and Shao, 2022) [14]. The employment of non-polluting and cleaner factors of production by new industries will lead to reduced pollutant emission, such as sulphur dioxide and soot, during industrial production (Chen, 2022) [16]. Specifically, the industrial upgrading index is calculated by assigning weights of 1, 2, and 3 to the primary, secondary and tertiary industries, respectively. A higher value indicates a more significant industrial upgrading (Zhang et al., 2023a) [10].

The findings in column (2) of Table 8 reveal that the coefficient of DE is significant and positive at the statistical level of 1%. Therefore, DE development proves to be an effective way to promote industrial structure upgrading. DE enables the reconfiguration and optimization of resource elements by virtue of the empowerment effect, scale effect, competition effect, squeezing effect, and substitution effect. With these advantages, it eliminates backward, low-end industries, accelerates environmental protection and scientific and technological development, promotes the transformation of enterprises towards digitalization, networking and intelligence, and facilitates green development and upgrading of related industries (Wu and Shao, 2022) [14]. Therefore, the industrial upgrading effect is an important mechanism for DE to promote urban EQ.

**5.3.3 Resource allocation effect.** Digital technology enhances the efficacy of resource usage and economic cycles by reconfiguring resource factors, like labor, capital and technology (Huong and Thanh, 2022) [9]. The reallocation and optimization of labor and other resources, along with the elimination of backward and low-end industries, will accelerate the industrialization of environmental protection scientific and technological advancements (Qi et al., 2022) [3]. In addition, technological advancements can alleviate the problem of uneven resource allocation resulting from information asymmetry and reduce production pollutant emission (Ma et al., 2023) [7]. According to Zhao et al. (2023a), resource allocation optimization utilizes total factor productivity, which is computed from urban employment and fixed asset investment, to depict the efficiency of resource allocation [13].

The findings in column (3) of Table 8 demonstrate that the estimated DE coefficient on resource allocation efficiency is 0.0128, which is a statistically significant result at the significance level of 10%. This shows that the development of DE can lead to the efficiency enhancement of resource allocation. By lowering search costs and reducing resource wastage, DE enhances the efficiency of resource allocation through optimized information sharing and price matching between supply and demand. The digital city governance model and technological innovation facilitated by DE can enhance resource allocation efficiency (Ma et al., 2023) [7]. Therefore, the allocation of resources is a crucial mechanism for DE to enhance the quality of urban environment.

**5.3.4 Environmental governance effect.** Environmental regulation involves the formulation of policies and measures by the government to regulate manufacturers and other economic activities that cause environmental pollution through the use of price and quantity tools (Hu et al., 2022) [17]. The aim is to reduce the external diseconomy caused by environmental pollution to an optimal level, and achieve sustainable development for both the environment and economy. Formal environmental regulation primarily originates from the government, as indicated by the proportion of investment in urban environmental governance (Yang et al., 2020) [46]. On the other hand, informal environmental regulation mainly stems from the public and environmental NGOs, as demonstrated by education levels and income (Yang and Liang, 2023) [47]. The three values are subsequently logarithmically averaged to characterize a city's total level of environmental regulation (Shen et al., 2023) [43].

The results in column (4) of Table 8 show that the estimated coefficient of DE on environmental regulation is 0.0026, which is statistically significant at the significance level of 1%. DE can enhance urban EQ through the effect of environmental governance. The Internet's openness, interactivity and real-time capabilities offer opportunities and convenience to the public (Liang and Yang, 2019) [4]. The Internet compensates for past shortcomings in environmental governance by increasing environmental intelligence and accuracy (Yang and Liang, 2023) [47]. Furthermore, the main body responsible for environmental governance evolves from single management to multiple coordination. This change contributes to a significant improvement in governance efficiency (Hu et al., 2023) [17]. As a result, the effectiveness of environmental regulation is a crucial tool for DE to enhance urban EQ.

**5.3.5 Infrastructure development effect.** China adopts strategies for constructing digital facilities, such as the Network Power Strategy, to bolster the healthy growth of DE (Zou and Pan, 2023) [8]. It is a crucial component of Chinese digital economic development plan. Broadband networks and other digital infrastructure greatly promote the green transformation of society and economy, which is conducive to the improvement of EQ (Zhang et al., 2023a) [10]. Digital infrastructure is measured using the ratio of employment in the information industry to employment in the tertiary sector (Huong and Thanh, 2022) [9].

The findings presented in column (5) of Table 8 indicate a significant positive correlation between DE and urban digital infrastructure at the statistical significance level of 1%. The

development of DG boosts the collaborative development of software and hardware application scenarios, such as artificial intelligence, data centers, cloud computing and smart healthcare, thereby promoting the development of strategic emerging industries and modern service industries, as well as the construction of new infrastructure based on digital technology (Qian et al, 2021; Li et al., 2022) [15, 28]. New infrastructure can utilize sensor technology, interconnected infrastructure, and other advanced devices to change the pattern of urban environmental governance and effectively improve urban EQ (Zhang et al., 2023a) [10]. Therefore, the development of urban infrastructure is an important path for DE to enhance urban EQ.

**5.3.6 Public habit effect.**   The demand for environmental sustainability amongst the public is a primary driving force behind environmental preservation (Liang and Yang, 2019) [4]. The improvement of online education, health and travel services, and the utilization of e-commerce and platform economies for consumption have a direct impact on the public cognitive behavior and lifestyle choices, ultimately aiding in fostering a green and low-carbon way of life (Chang et al., 2022) [19]. Public transport availability characterizes the living habits of the region population. Given that the transportation industry is a significant source of air pollutants and particulate emissions, we utilize travelling on public transport as a proxy for assessing the public environmentally conscious lifestyle (Chen et al., 2020) [48].

The results in column (6) of Table 8 show that the estimated coefficient of DE on urban public transportation is 0.0114, which is statistically significant at the significance level of 1%. The development of DE enhances urban EQ through a public habit effect. The use of digital technology via sharing economy, paper-free offices, online purchases, modern retail services, comprehensive one-stop offerings, teleconferencing, public transit, online learning, and the Internet + alters the public's community lifestyles to cater to individual demands while promoting eco-conscious consumption, resource conservation and a green and renewable way of living (Tang and Yang, 2023) [37]. Green public transport can enhance resource efficiency and reduce emission of pollutants, thereby improving ecological performance (Borck, 2019) [49].

Overall, the research indicates that DE development can enhance urban QE. This is achieved by promoting technological innovation, upgrading industrial structure, improving resource allocation, enhancing environmental governance, constructing environmental infrastructure, and fostering green living habits. These findings are in agreement with hypothesis 1.

## 5.4 Heterogeneity test

The influence of DE on urban EQ in China differs across cities based on their resource endowment, urban size and geographical location (Wang and Shi., 2022) [6]. To investigate the impact of DE on EQ of cities belonging to various types, this study analyses the effect of DE while considering a city's geographic location, administrative level, DE development policies, and environmental regulatory intensity.

**5.4.1 Geographic location.**   According to the criteria listed by the National Bureau of Statistics for the division of China's three major regions, the sample is divided into eastern, central and western cities. Columns (1) to (3) of Table 9 display the regression findings. DE in eastern and central cities has a significant positive impact on EQ. However, this impact is not significant in the western region, which is consistent with hypothesis H2a. Through the significance test of coefficient differences, it is found that the P-value of the regression coefficients for the eastern, central, and western regions was 0.036, indicating significant differences in the regression coefficients among the three regions. The reason lies in the more comprehensive digital infrastructure and environmental protection measures implemented in cities located in the east and central regions, as compared to those in the west. The western region lags in economic development, with traditional industries still accounting for a higher proportion of industrial

**Table 9. Heterogeneous effects of DE on the quality of the urban environment.**

|  | Urban Area | | | Administrative Level & Policy Support | | Type of Development | | Environmental Regulation | |
|---|---|---|---|---|---|---|---|---|---|
|  | Eas | Cen | Wes | Adv | Gen | Reso | Non-res | Str | Gen |
|  | (1) | (2) | (3) | (4) | (5) | (6) | (7) | (8) | (9) |
| DG | 0.031*** | 0.021*** | 0.007 | 0.022*** | 0.016** | 0.043*** | 0.029* | 0.024*** | 0.003 |
|  | (4.54) | (3.87) | (0.95) | (3.79) | (2.41) | (4.03) | (1.71) | (9.09) | (0.67) |
| Control variant | YES | YES | YES | YES | YES | YES | YES | YES | YES |
| City FE | YES | YES | YES | YES | YES | YES | YES | YES | YES |
| Year FE | YES | YES | YES | YES | YES | YES | YES | YES | YES |
| Obs | 1111 | 1100 | 924 | 374 | 2761 | 1265 | 1870 | 1859 | 1276 |
| P | 0.036** | | | 0.071* | | 0.059* | | 0.025** | |

Notes:

***, ** and * represent significance levels of 1%, 5% and 10% respectively; and values in brackets indicate t-values. The P is the result of the inter group coefficient difference test conducted using the Bootstrap method to extract samples 1000 times.

digitalization levels (Zou and Pan, 2023) [8]. Consequently, the level of digital industrialization is still at an early stage, with no scale effect yet established, leaving the effect of DE on urban EQ still in its infancy (Chen et al., 2020) [39].

**5.4.2 Administrative level and policy support for DE development.** The analysis of China's "White Paper" on smart cities and the division of big data comprehensive pilot zones indicate that pilot cities are mainly those with higher administrative levels in each province. Accordingly, this study categorizes pilot cities, municipalities, provincial capitals, and sub-provincial cities as high-level cities, while the remaining cities are classified as general cities, based on the list of China's smart city pilot units and national administrative levels. In general, high-level cities exhibit a greater level of development in DE compared to general cities. The results of the group regression analysis can be found in Column (4) and Column (5). DE of high-level cities can significantly enhance urban EQ at the significance level of 1%. However, the improvement effect of DE in general cities on urban EQ is only significant at the significance level of 5%. The P-value of the inter group coefficient difference also indicates that there is a remarkable difference in the direct regression coefficients between the two at the significance level of 10%, which conforms to research hypotheses H2b and H2c in this study. High-level cities benefiting from greater national policy concessions are particularly sensitive to DE development, and have more efficient policy attention and administrative order implementation. By contrast, general urban areas lack accumulated technology, capital, talent, and industrial foundation necessary for DE. As a result, their new digital infrastructure construction is relatively backward (Zou and Pan, 2023) [8]. Therefore, in cities with lower administrative levels, the improvement effect of DG on EQ is not as significant as that in higher-level cities. Undoubtedly, general cities need to accelerate their digital transformation.

**5.4.3 Type of urban development.** According to the "Development Plan for Resource-Based Cities (2013–2020)" issued by the State Council of China, the sample is categorized into 115 resource-based cities and 170 non-resource-based cities. Columns (6) and (7) in Table 9 present the test results for resource-based and non-resource-based cities respectively. DE enhances EQ of resource-based cities at the significance level of 1%, and only enhances EQ of non-resource-based cities at the significance level of 10%. The P-value of the inter group coefficients also proves the difference between the two groups of coefficients, which is consistent with research hypothesis H2d. The findings demonstrate that the proliferation of DE can not

only mitigate the reliance of resource-based cities on mineral resources and encourage the sustainable growth of the said cities, but also expedite the assimilation of novel technologies and business models in resource-based cities, thereby enhancing urban EQ. Furthermore, resource-based cities that rely on natural resources possess a delicate ecological environment (Tang and Wang, 2023) [37]. On the premise that the development level of DE is close, resource-based cities have greater potential to improve the ecological environment (Chen and Wang, 2022) [39].

**5.4.4 Intensity of urban environmental regulations.** This study establishes two categories of cities, namely key monitored cities and non-key monitored cities, based on the 169 cities listed by China's Ministry of Ecology and Environmental Protection for air quality monitoring. The regression findings in Column (8) and Column (9) demonstrate that digital economic growth can enhance the quality of urban environment in the nationally key monitored cities in terms of air quality, whereas it lacks a significant effect on the improvement of EQ in the non-monitored cities. The P-value of the inter group coefficients also proves the difference between the two groups of coefficients. This finding is in line with research hypothesis H2e. Due to the national environmental key monitoring list, urban environmental governance departments are now focusing on utilizing a range of digital technologies to efficiently monitor pollution. Implementing more advanced digital environmental governance strategies will lead to a more evident improvement in urban EQ (Zhao et al., 2023b) [25].

# 6. Conclusions and policy implications

## 6.1 Conclusions

This study explores the relationship between the level of DE and urban EQ in 285 prefecture-level cities in China. It examines the effect of DE on EQ and identifies the underlying mechanisms, using comprehensive measurements of both DE and urban EQ. This study draws the following main conclusions.

The findings of this study encompass several key aspects. Firstly, this study reveals that DE has a major beneficial impact on enhancing urban EQ. This assertion remains valid even after conducting robustness and endogeneity evaluations. Currently, DE lacks a spatial spillover effect on enhancing urban EQ, and the progression of regional DE does not result in the EQ enhancement of surrounding areas. Secondly, the impact paths indicate that DE can enhance urban EQ through the effects of technological innovation, upgrading of industrial structure, optimization of resource allocation, governmental environmental governance, and environmental governance as well as public living habit changes and infrastructure construction. Thirdly, the heterogeneity analysis reveals that the impact of DE on urban EQ is uneven and shows significant regional heterogeneity. Specifically, this effect is more pronounced in central and eastern cities, cities with higher administrative statuses, cities with supportive policies towards DE growth, resource-based cities, and cities with more stringent environmental regulations. This regional disparity can be attributed to differences in industrial structure, population distribution, policy support, economic level, and scientific and technological strength.

## 6.2 Policy implications

The current findings present compelling empirical evidence to support the promotion of China's innovation-driven strategy, the development of a "digital China", and the advancement of green economy. This study now proposes the following policy recommendations to achieve these goals.

Firstly, understanding the development trends and laws of DE is essential for enhancing the foundation of its healthy growth. Despite China's marked DE development, there are still areas

where improvement is needed. We should therefore adopt a strategy to strengthen our scientific and technological talent, enhance the policy support system for digital talent training, and continually boost the power and vitality of scientific and technological innovation. At the same time, a proper understanding of DE can enhance urban EQ by promoting the formulation and improvement of relevant policies that foster the development of digital infrastructure. Therefore, the digital infrastructure gap between western and eastern-central cities need bridging, thereby paving the road for a healthier growth of DE.

Secondly, improve the policy leadership of digital technological innovation whilst harnessing the green benefits of DE. Amidst the environmental constraints of DE, the government augments financial assistance for digital technology innovation through the refinement of the scientific and technological achievement reward system, thereby producing a collection of major milestones with fully autonomous intellectual property rights. By enhancing the safeguarding of scientific and technological accomplishments, the government reinforces the legal defense of intellectual property rights to overcome the obstacles posed by uncertain property boundaries and insufficient supervision of information security. It is vital to implement investment, tax, and subsidy policies that support early digital transformation of enterprises. Funding for R&D of key digital technologies should be continuously increased while promoting significant technological transformation in key fields and industries.

Thirdly, it is crucial to activate the central cities and city clusters as key players and enhance the regional cooperation mechanism to facilitate the ongoing development of DE. The governments can encourage in-depth collaboration between central cities and city clusters to drive DE development in non-central cities. This can be achieved by gradually transforming a central city into a visionary green metropolis led by DE, enabling it to effectively stimulate digital economic growth in both the central city and the surrounding city cluster. To enhance digital economic growth in urban areas, appropriate policy measures should be in place to support its development in underprivileged regions. This will prevent and combat the adverse effects of "digital islands", "data abuse" and "neighboring government competition" that could exacerbate existing problems.

Fourthly, it is imperative to establish a person-centered approach that considers the reasonable environmental demands and concerns of the public. As the guardians and stakeholders of urban EQ, residents, enterprises and local governments should engage in strategic decision-making regarding cost burden, interests and path choices. While considering the ecological requirements of inhabitants, businesses, and regional administrations, we ought to enhance environmental information sharing platforms and fine-tune the environmental regulatory framework. Public awareness of environmental protection can be heightened through guiding individuals toward establishing sustainable lifestyles and consumption habits. By stressing the importance of responsible decision-making and consumption patterns, we can collectively contribute to a more eco-friendly and sustainable future.

## Supporting information

**S1 Data.**
(DOCX)

## Acknowledgments

The authors would like to thank the reviewers and editors, as well as others who helped with the manuscript and whose suggestions greatly improved the manuscript.

## Author Contributions

**Data curation:** Hao Li, Zihan Yang.

**Formal analysis:** Hao Li.

**Methodology:** Hao Li.

**Software:** Hao Li.

**Visualization:** Hao Li.

**Writing – original draft:** Hao Li.

**Writing – review & editing:** Zihan Yang.

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
