## [Decision Letter · Decision Letter 0]

6 Sep 2023

PONE-D-23-23117An empirical study on the impact of digital economy development on urban environmental quality in ChinaPLOS ONE

Dear Dr. hao,

Thank you for submitting your manuscript to PLOS ONE. After careful consideration, we feel that it has merit but does not fully meet PLOS ONE’s publication criteria as it currently stands. Therefore, we invite you to submit a revised version of the manuscript that addresses the points raised during the review process.

We look forward to receiving your revised manuscript.

Kind regards,

Weike Zhang

Academic Editor

PLOS ONE

Journal Requirements:

   "The author(s) received no specific funding for this work.The funders had no role in study design, data collection and analysis, decision to publish, or preparation of the manuscript."

4. We note you have included a table to which you do not refer in the text of your manuscript. Please ensure that you refer to Table 2 and 3 in your text; if accepted, production will need this reference to link the reader to the Table.

Additional Editor Comments:

We have received two reports on your paper (Reviewer 1 reject and Reviewer 2 major revision) and, I have also read the paper myself. Please undertake the revision seriously and improve your writing following the style in this journal. Although I am giving you an opportunity to revise, I will be compelled to terminate the submission if the next version is found to be unsatisfactory. Overall, the reviewers are critical, particularly Reviwer 1, and the paper requires significant rewriting along with updating the literature.

1. In the introduction section, this part does not well written. This part should give why research this problem, how research, research results, etc. And also, authors should give the main contributions of this paper, and the differences with the other literature.

2. There are a lot of related paper published in the recent years, I want to know what the differences between the others. Authors should summary literature and find out the knowledge gaps. Besides, i suggest cite some recent papers from international journals. These papers maybe usefur for you to revise your manuscript. Does digital inclusive finance affect urban carbon emission intensity: Evidence from 285 cities in China. Cities. 142,11: 104552. https://doi.org/10.1016/j.cities.2023.104552; Reducing carbon emissions: Can high-speed railway contribute? Journal of Cleaner Production. 413, 8, 137524. ttps://doi.org/10.1016/j.jclepro.2023.137524；Corporate sustainable development driven by high-quality innovation: Does fiscal decentralization really matter? Economic Analysis and Policy. 78, 273-289.

3. In the empirical analysis, authors should give more economic analyses, instead of listing the regression results.

4.Authors need to check the full text.

5. I suggest authors carefully consider the suggestions of Reviewers 1 and 2

Reviewers' comments:

Reviewer's Responses to Questions

**Comments to the Author**

1. Is the manuscript technically sound, and do the data support the conclusions?

Reviewer #1: No

Reviewer #2: Partly

2. Has the statistical analysis been performed appropriately and rigorously? 

Reviewer #1: No

Reviewer #2: I Don't Know

3. Have the authors made all data underlying the findings in their manuscript fully available?

Reviewer #1: No

Reviewer #2: No

4. Is the manuscript presented in an intelligible fashion and written in standard English?

Reviewer #1: No

Reviewer #2: Yes

5. Review Comments to the Author

Reviewer #1: Multicollinearity leads to false results. There appears to be no link between the authors' theoretical analysis and empirical results, either in terms of baseline findings or regulatory mechanisms. In particular, the evaluation system constructed by the authors in Table 2 muddies the waters. Too many variables are not directly related to environmental quality, but are strongly related to selected control variables such as economic development and industrial structure, which leads to very severe multicollinearity. I don't want to dwell on these meaningless results.

Reviewer #2: The paper title "An empirical study on the impact of digital economy development on urban environmental quality in China" is a very comprehensive from very narrow area and cover each aspect of this area. The article analyse the relationship between digital economy and its effect on urban environmental quality in Chinese cities across the country. I recommend the paper for publication only if completed the guidelines of the journal and address the following queries/suggestions/comments.

1- I would suggest to rewrite the abstract into small understandable sentences. The abstract consist of 4-5 sentences. Some sentences are more than 4-5 lines.

2- Different arguments have been made in the introduction section, but most of those arguments don’t have any literature support except at one place the authors provide [1-10] references for one sentence without elaborating their contribution to science. Therefore, introduction part need serious attention.

3- Section 2, literature review; the authors tried divided the literature review into 8 sub headings (sub-sections) which doesn’t provide any foundation to form and argument as the objective of the paper. The authors need to write the literature review in proper sequence to give birth to their own arguments after providing relevant literature evidence.

4- Section 3: measurement of the level of development of digital economy……………… the authors provide Table 1 to present the level of digital economy development across cities. However, I can’t conclude the data from Table 1 last column. As some unit of measurements are %age, departments, kilometer, millions, billions, individuals, no of person etc etc. I don’t know how the authors form it into one model for estimation.

5- Similarly, Table 2 presents the data for urban environment quality. Again the unit of measurements for each index is different which again raises the concern that how the authors able to estimate it.

6- Section 3 does not seem to be very clear, especially since the author's introduction to data collection is rather vague. Since the quality of the data from any source is directly related to the reliability of the results and conclusions of the article, I suggest that the authors should first attach the data to the appendix, or at least let the reviewers read it to judge the quality of this data. Therefore, looking into the data is very necessary before validating these results.

7- Section 4.2, in the first few lines the statement shows the main variables followed by control variables. However, the authors didn’t focus on discussing the main variables and provide a good explanation for each control variable. I would suggest to do the same for main variables as well.

8- Table 6 provide the robustness and endogeneity tests. However, column 5 and 6 (tool variables) has no background discussion that where does it come from. Please discuss briefly what are these tool variables?

9- The conclusion of the paper need more discussion. The existing conclusion didn’t provide what has been done empirically.

10- Most of the reference in text are provided in numbers e.g. [1-10], [18] etc. However, some are presented in other format e.g. section 4.1 Dong (2018) and Xu (2022), also same problem with section 5.2.

6. PLOS authors have the option to publish the peer review history of their article (what does this mean?). If published, this will include your full peer review and any attached files.

Reviewer #1: No

Reviewer #2: No

---

## [Author Response · Author response to Decision Letter 0]

20 Oct 2023

Dear academic editor Zhang:

Thank you very much, Mr. Zhang, for the opportunity to revise this paper, especially the recent reference papers you provided. By carefully studying these papers, I have gained a better understanding of the standard writing paradigm of English academic papers and acquired more scientific research knowledge. In the future, I will continue to pay attention to Mr. Zhang's works and continue to learn from you.

In the revised draft of the article, in addition to revising the paper format, file naming, and financial information according to the style requirements of PLOS ONE, I have also revised the following issues raised by you. Specifically, these are as follows.

1. In the introduction section, this part does not well written. This part should give why research this problem, how research, research results, etc. And also, authors should give the main contributions of this paper, and the differences with the other literature.

2. There are a lot of related paper published in the recent years, I want to know what the differences between the others. Authors should summary literature and find out the knowledge gaps. Besides, i suggest cite some recent papers from international journals. These papers maybe usefur for you to revise your manuscript.

3. In the empirical analysis, authors should give more economic analyses, instead of listing the regression results.

4.Authors need to check the full text.

Response: After carefully studying the literature provided by teacher Zhang, i have followed the reference writing paradigm and rewritten why i am studying this paper, including the research method, research results, main contributions, and differences with other literature. I have also provided more economic explanations for the regression results and carefully reviewed the entire text. Many recent papers from international journals have been cited. Please refer to the red part of the revised manuscript for details.

The main gaps with the existing literature include:

In China, prior research has mostly relied on provincial data when examining the link between the digital economy and environmental pollution, rather than municipal data. Indeed, municipal data can better illustrate the contrast between digital economy development and environmental quality in Chinese cities than provincial data. Secondly, most of the existing literature focuses on analyzing the reduction of emissions caused by exhaust gas, wastewater discharge and solid waste. Given the complex interplay between urban environment quality and the digital economy, relying solely on a single-dimensional indicator may inadequately capture the multifaceted aspects of urban phenomena. Additionally, such an approach may introduce bias in empirical results and overlook the valuable purifying effect of urban natural ecology, as well as the human efforts invested in enhancing the overall quality of urban environment. It is therefore crucial to take a comprehensive approach when assessing urban environment quality. Thirdly, previous research has mainly focused on the overall effects of digital economic development on the urban environment or the spatial spillover effect between these two dimensions, providing only limited insight into the nonlinear, dynamic, and spatial spillover effects of digital economic development on the quality of the urban environment. Finally, while certain studies demonstrate the substantial impact of the digital economy on urban environmental quality, the involved underlying mechanisms remain ambiguous. Current research on digital economy development centers primarily on energy-saving, emission-reducing, technological innovation, and industrial upgrading effects, with less regard to the complete effects of digital economic growth on urban environmental quality. For instance, the advancement of digital technology in the economy has the potential to modify individual behavior, enhance environmental management, shift investment priorities, encourage urban infrastructure development and consequently alleviate ecological strain. 

Reviewer #1 Multicollinearity leads to false results. There appears to be no link between the authors' theoretical analysis and empirical results, either in terms of baseline findings or regulatory mechanisms. In particular, the evaluation system constructed by the authors in Table 2 muddies the waters. Too many variables are not directly related to environmental quality, but are strongly related to selected control variables such as economic development and industrial structure, which leads to very severe multicollinearity. I don't want to dwell on these meaningless results. 

Response: Thank you for your comments. In this article on the impact of the digital economy on urban environmental quality, we change the previous view that the development of the digital economy can directly affect the environment. The impact of the digital economy on the environment is reflected through indirect pathways, such as technological innovation, industrial upgrading, resource allocation, environmental governance, infrastructure construction and changes in public lifestyles. Therefore, after examining the relationship between the development of the digital economy and urban environmental quality in the empirical analysis, a comprehensive verification of the transmission path mechanism was conducted through the mediation effect model. Thus, there is a link between theoretical analyses and empirical results.

In the construction of the comprehensive indicator system, the development level of the digital economy referred to the construction method of the comprehensive indicator system for the development of the digital economy provided by the China Academy of Communications (2020). The comprehensive quality of the urban environment referred to the China Environmental Quality Comprehensive Evaluation Report written and published by Professor Yuan Xiaoling of Xi'an Jiaotong University in China. Since its first publication in 2013, four versions have been published. It provides a comprehensive evaluation of urban environmental quality from the perspective of the emission and absorption of urban environmental pollution.

In addition, during the comprehensive evaluation, various basic data were standardized and the dynamic comprehensive evaluation method (DCEM) was used to objectively assign weights to various indicators. The linear weighting method was then used to measure and calculate the comprehensive indicators. This index is a comprehensive indicator that has countless quantitative relationships with various basic indicators, so it will not have multiple collinear effects. Moreover, in the study, the author tried to adopt different characterization methods for the variables to be included in the equation model to avoid collinearity of the indicators as much as possible. Meanwhile, in the empirical analysis, the author also conducted multicollinearity tests on various variables and a series of robustness and endogeneity tests on the regression results. The test results indicate that there is no multicollinearity among the variables and the results are robust. 

Reviewer #2:

1-I would suggest to rewrite the abstract into small understandable sentences. The abstract consist of 4-5 sentences. Some sentences are more than 4-5 lines.

Response: Thank you for your comments and we agree with your suggestions. We have made corrections according to your comments. Please refer to the red part of the revised manuscript for details.

2- Different arguments have been made in the introduction section, but most of those arguments don’t have any literature support except at one place the authors provide [1-10] references for one sentence without elaborating their contribution to science. Therefore, introduction part need serious attention.

Response: Thank you for your comments and we agree with your suggestions. We have made corrections according to your comments. In the revised manuscript, the author corrected citations throughout the article and provided detailed information about the author, source, and contribution to the research in this article.

3- Section 2, literature review; the authors tried divided the literature review into 8 sub headings (sub-sections) which doesn’t provide any foundation to form and argument as the objective of the paper. The authors need to write the literature review in proper sequence to give birth to their own arguments after providing relevant literature evidence.

Response: Thank you for your comments. This article has reorganized the structure of the paper. In the introduction section, the literature on urban environmental quality, digital economy development and the relationship between digital economy development and urban environmental quality has been sorted out. The second section analyses the indirect path and heterogeneity factors of the impact of the digital economy on urban environmental quality. The specific modifications can be found in the original text.

4- Section 3: measurement of the level of development of digital economy, the authors provide Table 1 to present the level of digital economy development across cities. However, I can’t conclude the data from Table 1 last column. As some unit of measurements are %age, departments, kilometer, millions, billions, individuals, no of person etc etc. I don’t know how the authors form it into one model for estimation.

Response: Thank you for your sincere comments. This article mainly uses comprehensive evaluation indicators to comprehensively measure the development level of China urban digital economy and urban environmental quality. Among them, the development level of the digital economy refers to the construction method of the comprehensive indicator system for the development of the digital economy provided by the China Academy of Communications (2020). The comprehensive quality of urban environment mainly refers to the construction method in the China Environmental Quality Comprehensive Assessment Report prepared and published by Professor Yuan Xiaoling of Xi'an Jiaotong University in China. 

In the comprehensive measurement of comprehensive indicators, the first step is to standardize the basic data using the extreme value method. Second, the dynamic comprehensive evaluation method (DCEM) is used to objectively assign weights to each indicator to eliminate the influence between different units (see Guo, Y.J. Theory, Method and Application of Comprehensive Evaluation; China Science Publishing & Media: Beijing, China, 2007; ISBN 9787030187963.), and then use the linear weighting method to measure and calculate the comprehensive indicators. The main calculation steps are also presented in the revised manuscript of the article.

5- Similarly, Table 2 presents the data for urban environment quality. Again the unit of measurements for each index is different which again raises the concern that how the authors able to estimate it.

Response: Thank you for your comments. The approach to this issue is consistent with Question 4 and has been consistently presented in the answer to Question 4.

6- Section 3 does not seem to be very clear, especially since the author's introduction to data collection is rather vague. Since the quality of the data from any source is directly related to the reliability of the results and conclusions of the article, I suggest that the authors should first attach the data to the appendix, or at least let the reviewers read it to judge the quality of this data. Therefore, looking into the data is very necessary before validating these results.

Response: Thank you for your comments. In the new revised version, the authors have included the raw data used in the regression analyses in an appendix with details of the specific sources of each data. 

7- Section 4.2, in the first few lines the statement shows the main variables followed by control variables. However, the authors didn’t focus on discussing the main variables and provide a good explanation for each control variable. I would suggest to do the same for main variables as well.

Response: Thank you for your comments and we agree with your suggestions. In the new revised version, the authors provide a detailed description of the comprehensive indicator system of explanatory and interpretative variables, the measurement method and the results in Chapter III. In Chapter 4, the description of the variables, each control variable is described in detail.

8- Table 6 provide the robustness and endogeneity tests. However, column 5 and 6 (tool variables) has no background discussion that where does it come from. Please discuss briefly what are these tool variables?

Response: Thank you for your comments. In the new revised version, the authors detail the sources of instrumental variables in Chapter 4, Empirical Analysis Section, Endogeneity Tests. Please refer to the red part of the revised manuscript for details.

Considering the potential for biased regression outcomes stemming from endogeneity issues. This study employs the cross terms of historical data on post and telecommunications, represented by the number of post offices and telephones per 10,000 people at the end of 1984 in selected cities, together with previous year data on Internet users, as instrumental variables to determine the economic development levels of the cities (Wang et al., 2023). Digital technology originated from the promotion of fixed line telephone. Cities with high penetration of fixed line telephone in history generally have a high level of digital economic development. At the same time, before the popularization of fixed telephone, people used the post office system as the main way of information communication. The post office system is also the executive department of fixed telephone, which affects the construction of fixed telephone, Internet access, and other facilities to a certain extent; affects the development of digital economy, and meets the relevance. In addition, the impact of traditional telecommunications tools such as fixed line telephone and post office on current economic development is gradually reduced to meet exclusivity (Hu et al., 2023).

Wang H, Ding L, Guan R, Xia Y (2020) Effects of advancing internet technology on Chinese employment: a spatial study of interindustry spillovers. Technol Forecast Soc Chang 161:583–598. https://www.sciencedirect.com/science/article/abs/pii/S0040162520310854?via%3Dihub.

Hu J, Zhao X, Wu D, et al. Digital economy and environmental governance performance: Empirical evidence from 275 cities in China[J]. Environmental Science and Pollution Research, 2023, 30(10): 26012-26031. https://link.springer.com/article/10.1007/s11356-022-23646-w

9- The conclusion of the paper need more discussion. The existing conclusion didn’t provide what has been done empirically.

Response: Thank you for your sincere comments. Significant changes have been made to the content of this article, and the first part presents the differences between this article and existing research, as well as the main research contributions and conclusions. Please refer to the red part of the revised manuscript for details.

10- Most of the reference in text are provided in numbers e.g. [1-10], [18] etc. However, some are presented in other format e.g. section 4.1 Dong (2018) and Xu (2022), also same problem with section 5.2.

Response: Thank you for your comments and we agree with your suggestions. As suggested, the presentation of references in this paper has been standardized in the revised version, although some sites are presented slightly differently due to differences in presentation.

Last but not least, we would like to thank you again for your valuable comments on our manuscript. Thanks to these suggestions, the quality of this paper has been significantly improved. Of course, due to the limitation of personal ability, there may be still some flaws in the revised draft and you are welcome to point out the errors.

Best wishes

---

## [Decision Letter · Decision Letter 1]

16 Nov 2023

PONE-D-23-23117R1Does digital economy development affect urban environment quality: Evidence from 285 cities in ChinaPLOS ONE

Dear Dr. hao,

Thank you for submitting your manuscript to PLOS ONE. After careful consideration, we feel that it has merit but does not fully meet PLOS ONE’s publication criteria as it currently stands. Therefore, we invite you to submit a revised version of the manuscript that addresses the points raised during the review process.

We look forward to receiving your revised manuscript.

Kind regards,

Weike Zhang

Academic Editor

PLOS ONE

Additional Editor Comments:

We have received three reports on your revised paper (Reviewer 2 reject and Reviewer 3 and 4 major revisions) and, I have also read the paper myself. Please undertake the revision seriously and improve your writing following the style in this journal.

1. The introduction section does not well written. I recommend authors carefully revise this part follow some references in the top journals. You can refer to this paper: Seeing green: how does digital infrastructure affect carbon emission intensity? Energy Economics. 2023, 127, 107085. Of course, you can choose other relevant literature to refer to. For the literature provided by the reviewers or editor, you can choose whether it is suitable for this article.

Reviewers' comments:

Reviewer's Responses to Questions

**Comments to the Author**

1. If the authors have adequately addressed your comments raised in a previous round of review and you feel that this manuscript is now acceptable for publication, you may indicate that here to bypass the “Comments to the Author” section, enter your conflict of interest statement in the “Confidential to Editor” section, and submit your "Accept" recommendation.

Reviewer #2: (No Response)

Reviewer #3: (No Response)

Reviewer #4: All comments have been addressed

2. Is the manuscript technically sound, and do the data support the conclusions?

Reviewer #2: No

Reviewer #3: (No Response)

Reviewer #4: Partly

3. Has the statistical analysis been performed appropriately and rigorously? 

Reviewer #2: N/A

Reviewer #3: (No Response)

Reviewer #4: Yes

4. Have the authors made all data underlying the findings in their manuscript fully available?

Reviewer #2: Yes

Reviewer #3: (No Response)

Reviewer #4: Yes

5. Is the manuscript presented in an intelligible fashion and written in standard English?

Reviewer #2: No

Reviewer #3: (No Response)

Reviewer #4: No

6. Review Comments to the Author

Reviewer #2: The authors highlight most of the paper in red. But they didn't appropriately revised the paper as suggested. The ignore all the short comings of the paper through discussion in the author response section without incorporating it in the paper. I cannot recommend it for publication. Thank you

Reviewer #3: 1. The logical coherence of the introduction section needs improvement and further optimization is required.

2. It is recommended to maintain consistent terminology throughout the entire text, avoiding the interchange of "environmental pollution" and "environmental quality".

3. It is recommended to substitute the article by Li et al. (2021) with a superior quality one.

4. Why are there two titles in Table 2?

5. The definition and explanation of control variables are excessive; it is recommended to condense the content.

6. The coefficient test between groups is essential for conducting heterogeneity analysis in this study.

Reviewer #4: The authors have made adequate and careful revisions to answer the reviewer's concerns, but I think there are still some minor issues with the manuscript that need to be further improved.

1. The article discusses the topic of "digital economy" and "environmental quality" and analyzes the relationship between the two. In the introduction section, the manuscript only describes the relevant content of "environmental pollution", without the necessary elaboration on "environmental quality". The motivation and value of the article need to be further clarified.

2. In section 5.1, what are the methods and results of the multicollinearity test? The authors need to give a brief illustration, otherwise, it is not sufficient to convince the readers.

3. Regarding the analysis of the impact mechanisms, the authors transformed the control variables into mediating variables for the analysis. Is there a literature source or rationale for this treatment?

4. The manuscript needs to further improve the accuracy of its English presentation. E.g. "urban environment quality" appears in the title of the article, but "urban environmental quality" is used many times in the abstract and the main text. “Space Dubin” in Table 5 is supposed to be “Spatial Dubin”. Some other issues such as grammatical presentation need further revision.

5. Regarding the format of citation of literature, authors need to refer to the published papers of the journal and all the literature using numerical labeling format e.g. [1],[2-4].

7. PLOS authors have the option to publish the peer review history of their article (what does this mean?). If published, this will include your full peer review and any attached files.

Reviewer #2: No

Reviewer #3: No

Reviewer #4: No

---

## [Author Response · Author response to Decision Letter 1]

15 Dec 2023

Dear academic editor Zhang:

Thank you again for your valuable comments. Thank you for the literature you provide every time. This article has been revised according to the introduction writing style of the literature you provided. Every time I read the literature you provided, I have benefited greatly and deepened my knowledge of English writing. Reading your article made me more aware of my shortcomings. Every sentence and paragraph of your article is worth quoting. In the future, I will continue to pay close attention to your new work and study it carefully.

Reviewer #2: The authors highlight most of the paper in red. But they didn't appropriately revised the paper as suggested. The ignore all the short comings of the paper through discussion in the author response section without incorporating it in the paper. I cannot recommend it for publication. Thank you. 

Response: Thank you for your comments. I apologize that the revised paper still does not meet your expectations. I have addressed all of the reviewer's comments from the previous round and have not skipped any questions. However, it is possible that my answers were not precise enough to meet your needs. I will revise the paper once more and strive to obtain your approval. Most of the red marks on the thesis are due to significant content revisions, which may have made key issues less prominent. I apologize for any inconvenience caused.

Reviewer #3: 

1. The logical coherence of the introduction section needs improvement and further optimization is required.

Response: Thank you to the reviewer. The introductory paragraph of this article has been rewritten according to some relevant journal articles. Please refer to the original text and please point out if there are any shortcomings. I'm willing to continue making changes.

2- It is recommended to maintain consistent terminology throughout the entire text, avoiding the interchange of "environmental pollution" and "environmental quality".

Response: Thank you to the reviewer for identifying the issues. This paper builds upon previous research by scholars who have primarily focused on environmental pollution. However, we have expanded the research scope to include the overall environmental quality of the city as an explanatory variable. Therefore, the introduction may include some expressions related to environmental pollution. Your reminder helped to unify the research content by expressing it all in terms of urban environmental quality. To keep things concise, we will use the abbreviation EQ.

3- It is recommended to substitute the article by Li et al. (2021) with a superior quality one.

Response: Thank you for your comments. We have updated the research literature with superior quality articles. You can find them in the original article.

4- Why are there two titles in Table 2?.

Response: Apologies to the reviewer, here it was due to carelessness in writing, I have removed the redundant headings and rechecked the entire text. Thank you for your reminder.

5-The definition and explanation of control variables are excessive; it is recommended to condense the content.

Response: Thank you for bringing these issues to our attention. We have shortened the section on control variables as you recommended, and this can be seen in the revised version.

6- The coefficient test between groups is essential for conducting heterogeneity analysis in this study.

Response: According to your suggestion, this paper compared the size of regression coefficients and significance tests between the regression results of each group. Additionally, the Bootstrap method was used to take 1000 samples and test the significance of the coefficients. The original hypothesis was whether the difference of regression coefficients of each group is 0 or not. The test results' p-value shows a significant difference between the regression coefficients.

Reviewer #4: 1- The article discusses the topic of "digital economy" and "environmental quality" and analyzes the relationship between the two. In the introduction section, the manuscript only describes the relevant content of "environmental pollution", without the necessary elaboration on "environmental quality". The motivation and value of the article need to be further clarified.

Response: Thank you to the reviewer for identifying the issue. Previous studies have primarily focused on the impact of the digital economy on urban environmental pollution emissions. Building on this, this paper innovates by expanding the scope of the research to include the overall environmental quality of the city. By combining the city's environmental pollution emissions with urban environmental governance, nature absorption, and by constructing a comprehensive indicator system approach, the urban environmental quality is replaced by the urban environmental pollution. Using this method provides a broader and more meaningful research scope.

2- In section 5.1, what are the methods and results of the multicollinearity test? The authors need to give a brief illustration, otherwise, it is not sufficient to convince the readers.

Response: Thanks to the reviewer for pointing out the problems. The paper underwent a multicollinearity test using the Variance Inflation Factors (VIF) method. The results indicate that there is no multicollinearity between the variables as all VIF values are less than 10.

3- Regarding the analysis of the impact mechanisms, the authors transformed the control variables into mediating variables for the analysis. Is there a literature source or rationale for this treatment?

Response: Thank you for your sincere comments. While I have seen scholars approach this topic in a similar manner in the past, I apologize for not being able to locate that literature this time. To prevent inaccurate results caused by this approach, this paper includes technology level and industrial structure as control variables in the regression analysis, and the number of green patents and industrial structure upgrading as mediating variables. The technical level is measured by the natural logarithm of per capita financial expenditure on science and technology. The industrial structure is determined by the proportion of added value from the tertiary industries. The level of technological innovation is expressed by the number of green invention patents present in each region. The industrial upgrading index is calculated by assigning weights of 1, 2, and 3 to the primary, secondary, and tertiary industries, respectively. A higher value indicates a more significant industrial upgrading. The main findings did not change significantly after changing the variables.

4- The manuscript needs to further improve the accuracy of its English presentation. E.g. "urban environment quality" appears in the title of the article, but "urban environmental quality" is used many times in the abstract and the main text. “Space Dubin” in Table 5 is supposed to be “Spatial Dubin”. Some other issues such as grammatical presentation need further revision.

Response: Thank you for your comments. I apologize for any grammatical errors in my writing. The entire text has been thoroughly checked for grammatical and spelling errors, and a professional language editor has been hired to ensure that such mistakes are avoided in the future.

5-Regarding the format of citation of literature, authors need to refer to the published papers of the journal and all the literature using numerical labeling format e.g. [1],[2-4].

Response: Thank you for your comments. We agree with your suggestions. This paper has been written with reference to the formatting requirements of the journal, but as per your suggestion, I have used numbers like [1],[2-4] to indicate the format of the references.

We would like to thank you for your valuable comments on our manuscript. Your suggestions have significantly improved the quality of this paper. However, there may still be some flaws in the revised draft due to personal limitations. Please feel free to point out any errors.

Best wishes

---

## [Decision Letter · Decision Letter 2]

8 Jan 2024

Does digital economy development affect urban environment quality: Evidence from 285 cities in China

PONE-D-23-23117R2

Dear Dr. Li,

We’re pleased to inform you that your manuscript has been judged scientifically suitable for publication and will be formally accepted for publication once it meets all outstanding technical requirements.

Kind regards,

Weike Zhang

Academic Editor

PLOS ONE

Additional Editor Comments (optional):

Reviewers' comments:

Reviewer's Responses to Questions

**Comments to the Author**

1. If the authors have adequately addressed your comments raised in a previous round of review and you feel that this manuscript is now acceptable for publication, you may indicate that here to bypass the “Comments to the Author” section, enter your conflict of interest statement in the “Confidential to Editor” section, and submit your "Accept" recommendation.

Reviewer #4: All comments have been addressed

Reviewer #5: All comments have been addressed

2. Is the manuscript technically sound, and do the data support the conclusions?

Reviewer #4: Yes

Reviewer #5: Yes

3. Has the statistical analysis been performed appropriately and rigorously? 

Reviewer #4: Yes

Reviewer #5: Yes

4. Have the authors made all data underlying the findings in their manuscript fully available?

Reviewer #4: Yes

Reviewer #5: Yes

5. Is the manuscript presented in an intelligible fashion and written in standard English?

Reviewer #4: Yes

Reviewer #5: Yes

6. Review Comments to the Author

Reviewer #4: The authors have revised the manuscript and highlighted it in red. Their revisions have met my concerns. I recommend it for publication. Thank you.

Reviewer #5: The article examines the impact of the digital economy on urban environmental quality in China, focusing on 285 prefecture-level cities from 2011 to 2021. It finds a non-linear relationship and suggests that the digital economy positively impacts environmental quality, especially in central and eastern China and in cities with higher levels of administrative management and stringent environmental regulations and suggests that the digital economy positively impacts environmental quality, especially in central and eastern China and in cities with higher levels of administrative management and stringent environmental regulations. Here are my comments and suggestions:

1.The introduction requires improved logical coherence. It should clearly define the research scope and articulate the paper's objectives.

2.The paper should consistently use the term "urban environmental quality" (EQ) instead of interchanging it with "environmental pollution".

3.The manuscript requires further improvement in English language accuracy. For example, ensure consistency in terms like "urban environment quality" and correct typographical errors such as "Space Dubin" to "Spatial Durbin".

7. PLOS authors have the option to publish the peer review history of their article (what does this mean?). If published, this will include your full peer review and any attached files.

Reviewer #4: No

Reviewer #5: No

---

## [Editor Report · Acceptance letter]

12 Feb 2024

PONE-D-23-23117R2 

PLOS ONE

Dear Dr. Li, 

I'm pleased to inform you that your manuscript has been deemed suitable for publication in PLOS ONE. Congratulations! Your manuscript is now being handed over to our production team.

Kind regards, 

on behalf of

Dr. Weike Zhang 

Academic Editor

PLOS ONE